# Beyond Optimism:
# Exploration With Partially Observable Rewards

**Simone Parisi**
University of Alberta; Amii
parisi@ualberta.ca

**Alireza Kazemipour**
University of Alberta
kazemipo@ualberta.ca

**Michael Bowling**
University of Alberta; Amii
mbowling@ualberta.ca

## Abstract

Exploration in reinforcement learning (RL) remains an open challenge. RL algorithms rely on observing rewards to train the agent, and if informative rewards are sparse the agent learns slowly or may not learn at all. To improve exploration and reward discovery, popular algorithms rely on optimism. But what if sometimes rewards are *unobservable*, e.g., situations of partial monitoring in bandits and the recent formalism of monitored Markov decision process? In this case, optimism can lead to suboptimal behavior that does not explore further to collapse uncertainty. With this paper, we present a novel exploration strategy that overcomes the limitations of existing methods and guarantees convergence to an optimal policy even when rewards are not always observable. We further propose a collection of tabular environments for benchmarking exploration in RL (with and without unobservable rewards) and show that our method outperforms existing ones.

## 1 Introduction

Reinforcement learning (RL) has developed into a powerful paradigm where agents can tackle a variety of tasks, including games [67], robotics [31], and medical applications [82]. Agents trained with RL learn by trial-and-error interactions with an environment: the agent tries different actions, the environment returns a numerical reward, and the agent adapts its behavior to maximize future rewards. This setting poses a well-known dilemma: how and for how long should the agent explore, i.e., try new actions and visit new states in search for rewards? If the agent does not explore enough it may miss important rewards. However, prolonged exploration can be expensive, especially in real-world problems where instrumentation (e.g., cameras or sensors) or a human expert may be needed to provide rewards. Classic dithering exploration [39, 47] is known to be inefficient [34], especially when informative rewards are sparse [54, 60]. Among the many exploration strategies that have been proposed to improve RL efficiency, perhaps the most well-known and grounded is optimism. With optimism, the agent assigns to each state-action pair an optimistically biased estimate of future value and selects the action with the highest estimate [50]. This way, the agent is encouraged to explore unvisited states and perform different actions, where either its optimistic expectation holds — the observed reward matches the optimistic estimate — or not. If not, the agent adjusts its estimate and looks for rewards elsewhere. Optimistic algorithms range from count-based exploration bonuses [4, 16, 24, 25], to model-based planning [3, 20, 21], bootstrapping [15, 52], and posterior sampling [50]. Nonetheless, optimism can fail when the agent has limited observability of the outcome of its actions. Consider the example in Figure 1b, where the agent observes rewards only under some condition and upon paying a cost. To fully explore the environment and learn an optimal policy the agent must first take a *suboptimal* action (to push the button and pay a cost) to learn about the rewards. However, optimistic algorithms would never select suboptimal actions, thus never pushing the button and never learning how to accumulate positive rewards [36]. While alternatives to optimism exist, such as intrinsic motivation [66], they often lack guarantees of convergence and their efficacy strongly depends on the intrinsic reward used, the environment, and the task [7].

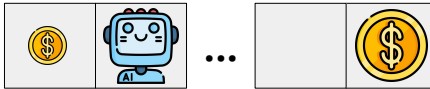 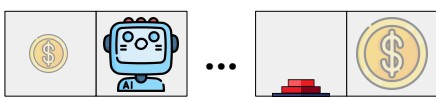

(a) Rewards given for collecting a coin are always observable.

(b) The agent must first push a button and pay a cost to observe coin rewards.

Figure 1: **When optimism is not enough.** The agent starts in the second leftmost cell of a corridor-like gridworld and can move LEFT or RIGHT. The coin in the leftmost cell gives a small reward, the one in the rightmost cell gives a large reward, while all other cells give zero rewards. In (b), pushing the button is costly (negative reward) but is needed to *observe* coin rewards — if the agent collects a coin without pushing it first, *the environment returns the reward but the agent cannot observe it*. Given a sufficiently large discount factor, the optimal policy is to collect the large coin without pushing the button. Consider a purely optimistic agent, i.e., an agent that selects action greedily with respect to their value estimate, and these estimates are initialized to the same optimistically high value [17]. In (a) all rewards are always observable, therefore: the agent visits a cell; its optimistic estimate decreases according to the reward; the agent tries another cell. At the end, it will visit all states and learn the optimal policy. In (b), however, when the agent visits a coin cell *without pushing the button first, it will not observe the reward and will not validate or prove wrong its optimistic estimate*. Thus, the estimate for both coin-cells will stay equally optimistic, and between the two the agent will prefer to go to the leftmost cell because it is closer to the start. Optimistic model-based algorithms [3, 24] have the same problem. If all value estimates are optimistic and the agent knows that pushing the button is costly, between (1) push the button and collect the right coin, (2) do not push the button and collect the right coin, and (3) do not push the button and collect the left coin, the agent will always chose the third: the optimistic value is the same, but it does not incur in the button cost and the left coin is closer. Yet, this gives no information to the agent, because it cannot observe the reward and therefore cannot update its optimistic value. Note that the agent could replace the unobserved reward with an estimate from a model updated as rewards are observed. But how can this model be accurate if the agent cannot explore properly and observe rewards in the first place?

*How can we efficiently explore and learn to act optimally when rewards are partially observable, without relying on optimism and yet still have guarantees of convergence?* In this paper, we present a novel exploration strategy based on the successor representation to tackle this question. Note that Lattimore and Szepesvari [36] already argued against optimism in partial monitoring [8], a generalization of the bandit framework where the agent cannot observe the exact payoffs. Similarly, Schulze and Evans [68] and Krueger et al. [33] discussed the shortcomings of optimism in active RL, where rewards are observable upon paying a cost. While we follow the same line of research, we consider the more recent and general framework of Monitored Markov Decision Processes (Mon-MDPs) [55], and we validate the efficacy of the proposed exploration on a collection of both MDPs and Mon-MDPs.

## 2 Problem Formulation

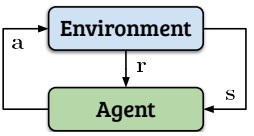

Figure 2: **The MDP framework.**

A Markov Decision Process (MDP) (Figure 2) is a mathematical framework for sequential decision-making, defined by the tuple $\langle \mathcal{S}, \mathcal{A}, \mathcal{R}, \mathcal{P}, \gamma \rangle$. An agent interacts with an environment by repeatedly observing a state $s_t \in \mathcal{S}$, taking an action $a_t \in \mathcal{A}$, and observing a bounded reward $r_t \in \mathbb{R}$. These interactions are governed by the Markovian transition function $\mathcal{P}(s_{t+1} | a_t, s_t)$ and the reward function $r_t \sim \mathcal{R}(s_t, a_t)$, both unknown to the agent. The agent's goal is to act to maximize the sum of discounted rewards $\sum_{t=1}^{\infty} \gamma^{t-1} r_t$, where $\gamma \in [0, 1)$ is the discount factor describing the trade-off between immediate and future rewards.

How the agent acts is determined by the *policy* $\pi(a | s)$, a function denoting the probability of executing action $a$ in state $s$. A policy is optimal when it maximizes the Q-function, i.e.,

$$\pi^* := \arg\max_{\pi} Q^{\pi}(s, a), \qquad \text{where } Q^{\pi}(s_t, a_t) := \mathbb{E}\Big[\sum_{k=t}^{\infty} \gamma^{k-t} r_k \,|\, \pi, \mathcal{P}, s_t, a_t\Big]. \qquad (1)$$

The existence of at least one optimal policy and its optimal Q-function $Q^*$ is guaranteed under the following standard assumptions: finite state and action spaces, stationary reward and transition functions, and bounded rewards [58]. Q-Learning [81] is one of the most common algorithms to

learn $Q^*$ by iteratively updating a function $\widehat{Q}$ as follows,

$$\widehat{Q}(s_t, a_t) \leftarrow (1 - \alpha_t)\widehat{Q}(s_t, a_t) + \alpha_t(r_t + \gamma \overbrace{\max_a \widehat{Q}(s_{t+1}, a)}^{\text{greedy policy}}), \qquad (2)$$

where $\alpha_t$ is the learning rate. $\widehat{Q}$ is guaranteed to converge to $Q^*$ under an appropriate learning rate schedule $\alpha_t$ and if every state-action pair is visited infinitely often [13]. Thus, how the agent explores plays a crucial role. For example, a random policy eventually visits all states but is clearly inefficient, as states that are hard to reach will be visited less frequently (e.g., because they can be reached only by performing a sequence of specific actions). How frequently the policy explores is also crucial — to behave optimally we also need the policy to act optimally eventually. That is, at some point, the agent should stop exploring and act greedy instead. For these reasons, we are interested in *greedy in the limit with infinite exploration (GLIE)* policies [69], i.e., policies that guarantee to explore all state-action pairs enough for $\widehat{Q}$ to converge to $Q^*$, and that eventually act greedy as in Eq. (1). While a large body of work in RL literature has been devoted to developing efficient GLIE policies in MDPs, an important problem still remains largely unaddressed: *the agent explores in search of rewards, but what if rewards are not always observable?* MDPs, in fact, assume rewards are always observable. Thus, we consider the more general framework of Monitored MDPs (Mon-MDPs) [55].

## 2.1 Monitored MDPs: When Rewards Are Not Always Observable

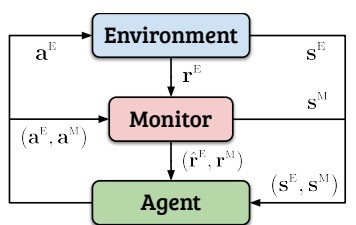

Figure 3: **The Mon-MDP framework.**

Mon-MDPs (Figure 3) are defined by the tuple $\langle \mathcal{S}^{\text{E}}, \mathcal{A}^{\text{E}}, \mathcal{P}^{\text{E}}, \mathcal{R}^{\text{E}}, \mathcal{M}, \mathcal{S}^{\text{M}}, \mathcal{A}^{\text{M}}, \mathcal{P}^{\text{M}}, \mathcal{R}^{\text{M}}, \gamma \rangle$, where $\langle \mathcal{S}^{\text{E}}, \mathcal{A}^{\text{E}}, \mathcal{P}^{\text{E}}, \mathcal{R}^{\text{E}}, \gamma \rangle$ is the same as classic MDPs and the superscript E stands for "environment", while $\langle \mathcal{M}, \mathcal{S}^{\text{M}}, \mathcal{A}^{\text{M}}, \mathcal{P}^{\text{M}}, \mathcal{R}^{\text{M}} \rangle$ is the "monitor", a separate process with its own states, actions, rewards, and transition. The monitor works similarly to the environment — performing a monitor action $a_t^{\text{M}} \in \mathcal{A}^{\text{M}}$ affects its state $s_t^{\text{M}} \in \mathcal{S}^{\text{M}}$ and yields a bounded reward $r_t^{\text{M}} \sim \mathcal{R}^{\text{M}}(s_t^{\text{M}}, a_t^{\text{M}})$. However, there are two crucial differences. First, its Markovian transition depends on the environment as well, i.e., $\mathcal{P}^{\text{M}}(s_{t+1}^{\text{M}} | s_t^{\text{M}}, s_t^{\text{E}}, a_t^{\text{M}}, a_t^{\text{E}})$. Second, the *monitor function* $\mathcal{M}$ stands in between the agent and the environment, dictating what the agent sees about the environment reward — rather than directly observing the *environment reward* $r_t^{\text{E}} \sim \mathcal{R}^{\text{E}}(s_t^{\text{E}}, a_t^{\text{E}})$, the agent observes a *proxy reward* $\hat{r}_t^{\text{E}} \sim \mathcal{M}(r_t^{\text{E}}, s_t^{\text{M}}, a_t^{\text{M}})$. Most importantly, the monitor may not always show a numerical reward, and the agent may sometime receive $\hat{r}_t^{\text{E}} = \perp$, i.e., *"unobservable reward"*.

In Mon-MDPs, the agent repeatedly executes a joint action $a_t := (a_t^{\text{E}}, a_t^{\text{M}})$ according to the joint state $s_t := (s_t^{\text{E}}, s_t^{\text{M}})$. In turn, the environment and monitor states change and produce a joint reward $(r_t^{\text{E}}, r_t^{\text{M}})$, but the agent observes $(\hat{r}_t^{\text{E}}, r_t^{\text{M}})$. The agent's goal is to select joint actions to maximize $\sum_{t=1}^{\infty} \gamma^{t-1} (r_t^{\text{E}} + r_t^{\text{M}})$ *even though it observes $\hat{r}_t^{\text{E}}$ instead of $r_t^{\text{E}}$*. For example, in Figure 1b the agent can turn monitoring on/off by pushing the button: if monitoring is off, the agent cannot observe the environment rewards; if on, it observes them but receives negative monitor rewards. Formally, $\mathcal{S}^{\text{M}} = \{\text{ON}, \text{OFF}\}$, $\mathcal{A}^{\text{M}} = \{\text{PUSH}, \text{NO-OP}\}$, $\mathcal{R}^{\text{M}}(\text{ON}, \cdot) = -1$, $\mathcal{M}(r_t^{\text{E}}, \text{ON}, \cdot) = r_t^{\text{E}}$, $\mathcal{M}(r_t^{\text{E}}, \text{OFF}, \cdot) = \perp$. The optimal policy is to move to the rightmost cell without turning monitoring on — the agent still receives the positive reward even though it does not observe it.[1]

The presence of the monitor and the partial observability of rewards highlight two fundamental differences between Mon-MDPs and existing MDPs variations. First, *the observability of the reward is dictated by a "Markovian entity" (the monitor)*, thus actions can have either immediate or long-term effects on the reward observability. For example, the agent may immediately observe the reward upon explicitly asking for it as in active RL [68], e.g., with a dedicated action $a^{\text{M}} = \text{ASK}$. Or, rewards may become observable only after changing the monitor state through a sequence of actions, e.g., as in Figure 1b by reaching and pushing the button. Second, *reward unobservability goes beyond reward sparsity*. In sparse-reward MDPs [34], rewards are *always* observable even though informative non-zero rewards are sparse. Mon-MDPs with dense informative observable rewards may still be challenging even if a few rewards are partially observable. Consider the Mon-MDP in Figure 1b, but

---

[1]This example is just illustrative. In general, the monitor states and actions can be more heterogeneous. For example, the agent could observe rewards by collecting and using a device or asking a human expert, and the monitor state could include the battery of the device or the location of the expert. Similarly, the monitor reward is not constrained to be negative and its design depends on the agent's desired behavior.

this time zero-rewards of empty tiles are replaced with negative rewards proportional to the distance to the large coin (dense and informative). Because coin rewards are still observable only after pressing the button, an optimistic agent will still collect the small coin: the optimistic value of the small coin is as high as the optimistic value of the large coin, but collecting the small coin takes fewer steps, thus ending the series of dense negative rewards sooner. *The agent must first push the button (a suboptimal action) to learn the optimal policy regardless of the presence of dense observable rewards.*

Mon-MDPs provide a comprehensive framework encompassing existing areas of research, such as active RL [33, 68] and partial monitoring [5, 8, 37, 41]. In their introductory work, Parisi et al. [55] showed the existence of an optimal policy and the convergence of a variant of Q-Learning under some assumptions.[2] Yet, many research questions remain open, most prominently how to explore efficiently in Mon-MDPs. In the next section, we review exploration strategies in RL and discuss their shortcomings, especially when rewards are partially observable as in Mon-MDPs. For the sake of simplicity, in the remainder of the paper we use the notation $(s, a)$ to refer to both the environment state-action pairs (in MDPs) and the *joint* environment-monitor state-action pairs (in Mon-MPDs). All the related work on exploration for MDPs can be applied to Mon-MDPs as well (not albeit with limited efficacy, as we discuss below). Similarly, the exploration strategy we propose in Section 3 can be applied to both MDPs and Mon-MDPs.

## 2.2 Related Work

**Optimism.** In tabular MDPs, many provably efficient algorithms are based on optimism in the face of uncertainty (OFU) [35]. In OFU, the agent acts greedily with respect to an optimistic estimate composed of the action-value estimate (e.g., $\widehat{Q}$) and a bonus term that is proportional to the uncertainty of the estimate. After executing an optimistic action, the reward either validates the agent's optimism or proves it wrong, thus discouraging the agent from trying it again. RL literature provides many variations of these algorithms, each with different convergence bounds and assumptions. Perhaps the best known OFU algorithms are based on the upper confidence bound (UCB) algorithm [4], where the optimistic value is computed from visitation counts (the lower the count, the higher the bonus). In model-free RL, Jin et al. [25] proved convergence bounds for episodic MDPs, and later Dong et al. [16] extended the proof to infinite horizon MDPs, but both are often intractable. In model-based RL, algorithms like R-MAX [10] and UCRL [3, 24] consider a set of plausible MDP models (i.e., plausible reward and transition functions) and optimistically select the best action over what they believe to be the MDP yielding the highest return. Unfortunately, this optimistic selection can fail when rewards are not always unobservable [33, 36, 68], as the agent may have to select actions that are *known* to be suboptimal to discover truly optimal actions. For example, in Figure 1b pushing the button is suboptimal (the agent pays a cost) but is the only way to discover the large positive reward.

**Posterior sampling for RL (PSRL).** PSRL adapts Thompson sampling [79] to RL. The agent is given a prior distribution over a set of plausible MDPs, and rather than sampling the MDP optimistically as in OFU, PSRL samples it from a distribution representing how likely an MDP is to yield the highest return. Only then, the agent acts optimally for the sampled MDP. PSRL can be orders of magnitude more efficient than UCRL [50], but its efficiency strongly depends on the choice of the prior [32, 57, 73, 80]. Furthermore, the computation of the posterior (by Bayes' theorem) can be intractable even for small tasks, thus algorithms usually approximate it empirically with an ensemble of randomized Q-functions [15, 51, 52]. Nonetheless, if rewards are partially observable PSRL suffers from the same problem as OFU: the agent may never discover rewards if actions needed to observe them (and further identify the MDP) are suboptimal for all MDPs with posterior support [36].

**Information gain.** These algorithms combine PSRL and UCB principles. On one hand, they rely on a posterior distribution over plausible MDPs, on the other hand they augment the current value estimate with a bound that comes in terms of an information gain quantity, e.g., the difference between the entropy of the posterior after an update [28, 40, 62, 70]. This allows to sample suboptimal but informative actions, thus overcoming some limitations of OFU and PSRL. However, similar to PSRL, performing explicit belief inference can be intractable even for small problems [2, 29, 62].

**Intrinsic motivation.** While many of above strategies are often computationally intractable, they

---

[2]The assumptions are: *monitor and environment ergodicity*, i.e., any joint state-action pair can be reached by any other pair given infinite exploration; *monitor ergodicity*, i.e., for every environment reward there exists at least one joint state-action pair for which the environment reward is observable given infinite exploration; *truthful monitor*, i.e., either the proxy reward is the environment reward ($\hat{r}_t^{\text{E}} = r_t^{\text{E}}$) or is not observable ($\hat{r}_t^{\text{E}} = \perp$).

inspired more practical algorithms where exploration is conducted by adding an *intrinsic reward* to the environment. This is often referred to as intrinsic motivation [63, 66]. For example, the intrinsic reward can depend on the visitation count [9], the impact of actions on the environment [60], interest or surprise [1, 53], information gain [22], or can be computed from prediction errors of some quantity [11, 56, 71]. However, intrinsic rewards introduce non-stationarity to the Q-function (they change over time), are often myopic (they are often computed only on the immediate transition), and are sensitive to noise [56]. Furthermore, if they decay too quickly exploration may vanish prematurely, but if they outweigh the environment reward the agent may explore for too long [11].

It should be clear that learning with partially observable rewards is still an open challenge. In the next section, we present a practical algorithm that (1) performs deep exploration — it can reason long-term, e.g., it tries to visit states far away, (2) does not rely on optimism — addressing the challenges of partially observable rewards, (3) explicitly decouples exploration and exploitation — thus exploration does not depend on the accuracy of $\widehat{Q}$, which is known to be problematic (e.g., overestimation bias) for off-policy algorithms (even more if rewards are partially observable). After proving its asymptotic convergence we empirically validate it on a collection of tabular MDPs and Mon-MDPs.

## 3 Directed Exploration-Exploitation With The Successor Representation

---

**Algorithm 1:** Directed Exploration-Exploitation

---

1 $(s^{\mathrm{G}}, a^{\mathrm{G}}) = \arg\min_{s,a} N_t(s, a)$        // tie-break by deterministic ordering
2 $\beta_t = {}^{\log t}/_{N_t(s^{\mathrm{G}}, a^{\mathrm{G}})}$
3 **if** $\beta_t > \bar{\beta}$ **then return** $\rho(a \,|\, s_t, s^{\mathrm{G}}, a^{\mathrm{G}})$        // explore
4 **else return** $\arg\max_a \widehat{Q}(s_t, a)$        // exploit

---

Algorithm 1 outlines our idea for step-wise exploration.[3] $N(s, a)$ is the state-action visitation count and, at every timestep, the agent selects the least-visited pair as "goal" $(s^{\mathrm{G}}, a^{\mathrm{G}})$. The ratio $\beta_t$ decides if the agent explores (goes to the goal) or exploits according to a threshold $\bar{\beta} > 0$. If two or more pairs have the same lowest count, ties are broken consistently according to some deterministic ordering. For example, if actions are $\{\texttt{LEFT}, \texttt{RIGHT}, \texttt{UP}, \texttt{DOWN}\}$, $\texttt{LEFT}$ is the first and will be selected if the count of all actions is the same. This ensures that once the agent starts exploring to reach a goal, it will continue to do so until the goal is visited — once its count is minimal under the deterministic tie-break, it will continue to be minimal until the goal is visited and its count is incremented. Exploration is driven by $\rho(a \,|\, s_t, s^{\mathrm{G}}, a^{\mathrm{G}})$, a *goal-conditioned policy* that selects the action according to the current state $s_t$ and goal $(s^{\mathrm{G}}, a^{\mathrm{G}})$. Intuitively, as the agent explores, if every state-action pair is visited infinitely often, $\log t$ will grow at a slower rate than $N_t(s, a)$ and the agent will increasingly often exploit (formal proof below). Furthermore, infinite exploration ensures $\widehat{Q}$ converges to $Q^*$, i.e., that its greedy actions are optimal. Intuitively, the role of $\beta_t$ is similar to the one of $\epsilon_t$ in $\epsilon$-greedy policies [75]. However, rather than manually tuning its decay, $\beta_t$ naturally decays as the agent explores.

The goal-conditioned policy $\rho$ is the core of Algorithm 1. For efficient exploration we want $\rho$ to move the agent as quickly as possible to the goal, i.e., we want to minimize the *goal-relative diameter* under $\rho$. Intuitively, this diameter is a bound on the expected number of steps the policy would take to visit the goal from any state (formal definition below). Moreover, we want to untie $\rho$ from the Q-function, the rewards, and their observability. In the next section we propose a policy $\rho$ that satisfies these criteria, but first we prove that Algorithm 1 is a GLIE policy under some assumptions on $\rho$.

**Definition 1** (Singh et al. [69])**.** *An exploration policy is greedy in the limit (GLIE) if (1) each action is executed infinitely often in every state that is visited infinitely often, and (2) in the limit, the learning policy is greedy with respect to the Q-function with probability 1.*

**Definition 2.** *Let $\rho$ be a goal-conditioned policy $\rho(a \,|\, s, s^{\mathrm{G}}, a^{\mathrm{G}})$. Let $T(s^{\mathrm{G}}, a^{\mathrm{G}} \,|\, \rho, s)$ be the first timestep $t$ in which $(s_t, a_t) = (s^{\mathrm{G}}, a^{\mathrm{G}})$ given $s_0 = s$ and $a_t \sim \rho(a \,|\, s_t, s^{\mathrm{G}}, a^{\mathrm{G}})$. The goal-relative diameter of $\rho$ with respect to $(s^{\mathrm{G}}, a^{\mathrm{G}})$, if it exists, is*

$$D^\rho(s^{\mathrm{G}}, a^{\mathrm{G}}) = \max_s \mathbb{E}[T(s^{\mathrm{G}}, a^{\mathrm{G}} \,|\, \rho, s) \,|\, \mathcal{P}, \rho], \tag{3}$$

*i.e., the maximum expected number of steps to reach $(s^{\mathrm{G}}, a^{\mathrm{G}})$ from any state in the MDP. We say the goal-relative diameter is bounded by $\bar{D}$ if $\bar{D} \geq \max_{s^{\mathrm{G}}, a^{\mathrm{G}}} D^\rho(s^{\mathrm{G}}, a^{\mathrm{G}})$.*

---

[3]Recall that $s$ and $a$ denote either the classic state and action in MDPs, or the joint state $s := (s^{\mathrm{E}}, s^{\mathrm{M}})$ and action $a := (a^{\mathrm{E}}, a^{\mathrm{M}})$ in Mon-MDPs.

There exist goal-conditioned policies with bounded goal-relative diameter if and only if the MDP is communicating (i.e., for any two states there is a policy under which there is non-zero probability to move between the states in finite time; see Puterman [58] and Jaksch et al. [24]). While not all goal-conditioned policies have bounded goal-relative diameter, one such policy is the random policy or similarly an $\epsilon$-greedy policy for $\epsilon > 0$ [69]. Different goal-conditioned policies will have different bounds on their goal-relative diameter $D^\rho(s^G, a^G)$.[4]

**Theorem 1.** *If the goal-relative diameter of $\rho$ is bounded by $\bar{D}$ then Algorithm 1 is a GLIE policy.*

*Proof.* Let $Z_t(s,a)$ be the number of timesteps before time $t$ that the agent is in an exploration step with $(s^G, a^G) = (s, a)$. Let $X_t = \sum_{s,a} Z_t(s,a)/t$ be the fraction of time up to time $t$ that the agent has spent exploring. We want to show that $\forall \varepsilon > 0 \lim_{t \to \infty} \Pr[X_t > \varepsilon] = 0$, i.e., the probability that the agent explores as frequently as any positive $\varepsilon$ approaches 0. Hence, the policy is greedy in the limit. Note that, if the agent's frequency of exploring approaches zero this also must mean that $\beta_t \in \mathcal{O}(1)$, implying $N_t(s,a) \to \infty \ \forall(s,a)$, i.e., all state-action pairs are visited infinitely often.

Let us focus on $\mathbb{E}[Z_t(s,a)]$. Once the agent starts exploring to visit $(s,a)$, it will do so until it actually visits $(s,a)$. Let $I_t(s,a)$ be the number of times the agent started exploring to visit $(s,a)$ before time $t$. We know that $I_t(s,a) \leq N_t(s,a) + 1$, as the agent must visit $(s,a)$ before it starts exploring to visit it again. Let $t' \leq t$ be the last time it started exploring to visit $(s,a)$. We have

$$I_t(s,a) = I_{t'}(s,a) \leq N_{t'}(s,a) + 1 < \frac{\log t'}{\bar{\beta}} + 1 \leq \frac{\log t}{\bar{\beta}} + 1, \tag{4}$$

where the strict inequality is due to $\beta_{t'} > \bar{\beta}$ (condition to explore). Since the agent cannot have started exploration $\log t/\bar{\beta} + 1$ times, and the goal-relative diameter of $\rho$ is at most $\bar{D}$, then

$$\mathbb{E}[Z_t(s,a)] < \bar{D}\left(\frac{\log t}{\bar{\beta}} + 1\right), \quad \text{and thus} \quad \mathbb{E}[X_t] < \frac{|\mathcal{S}||\mathcal{A}|\bar{D}}{t}\left(\frac{\log t}{\bar{\beta}} + 1\right). \tag{5}$$

We can now apply Markov's inequality with threshold $1/\sqrt{t}$,

$$\Pr\left[X_t \geq \frac{1}{\sqrt{t}}\right] \leq \sqrt{t}\,\mathbb{E}[X_t] < \frac{|\mathcal{S}||\mathcal{A}|\bar{D}}{\sqrt{t}}\left(\frac{\log t}{\bar{\beta}} + 1\right). \tag{6}$$

Since $1/\sqrt{t} < \varepsilon$ for sufficiently large $t$,

$$\lim_{t \to \infty} \Pr[X_t \geq \varepsilon] \leq \lim_{t \to \infty} \Pr\left[X_t \geq \frac{1}{\sqrt{t}}\right] \leq \lim_{t \to \infty} \frac{|\mathcal{S}||\mathcal{A}|\bar{D}}{\sqrt{t}}\left(\frac{\log t}{\bar{\beta}} + 1\right) = 0, \tag{7}$$

hence the Algorithm 1 is a GLIE policy. $\qquad \square$

**Corollary 1.** *As a consequence of Theorem 1, $\widehat{Q}$ converges to $Q^*$ (infinite exploration) and therefore the algorithm's behavior converges to the optimal policy (greedy in the limit).[5]*

While we have noted that the random policy is a sufficient choice for $\rho$ to meet the criteria of Theorem 1, the bound in Equation 7 shows a direct dependence on the goal-relative diameter $D^\rho(s^G, a^G)$. Thus, the optimal $\rho$ to explore efficiently in Algorithm 1 is $\rho^*(a \,|\, s, s^G, a^G) := \arg\min_\rho D^\rho(s^G, a^G)$, i.e., we want to reach the goal from any state as quickly as possible in expectation. Furthermore, we want to untie $\rho$ from the learned Q-function and the reward observability. However, learning such a policy can be challenging because the diameter of the MDP is usually unknown. In the next section, we present a suitable alternative based on the successor representation [14].

### 3.1 Successor-Function: An Exploration Compass

*The successor representation (SR)* [14] is a generalization of the value function, and represents the cumulative discounted occurrence of a state $s_i$ under a policy $\pi$, i.e., $\mathbb{E}[\sum_{k=t}^{\infty} \gamma^{k-t} \mathbb{1}_{\{s_k=s_i\}} \,|\, \pi, \mathcal{P}, s_t]$, where $\mathbb{1}_{\{s_k=s_i\}}$ is the indicator function returning 1 if $s_k = s_i$ and 0 otherwise. The SR does not depend on the reward function and can be learned with temporal-difference in a model-free fashion, e.g., with Q-Learning. Despite their popularity in transfer RL [6, 19, 38, 61], to the best of our knowledge, only Machado et al. [43] and Jin et al. [26] used the SR to enhance exploration by using it as an intrinsic reward. Here, we follow the idea of the SR — to predict state occurrences under

---

[4]Jaksch et al. [24] defined a diameter that is a property of the MDP irrespective of any (goal-directed) policy. Their MDP diameter can be thought of as the smallest possible bound on the goal-relative diameter for any $\rho$.

[5]For Mon-MDPs, convergence is guaranteed under the assumptions discussed in Footnote [2].

a policy — to learn a value function that the goal-conditioned policy in Algorithm 1 can use to guide the agent towards desired state-action pairs. We call this the *state-action successor-function* (or S-function) and we formalize it as a general value function [76] representing the cumulative discounted occurrence of a state-action pair $(s_i, a_j)$ under a policy $\pi$, i.e.,

$$S_{s_i a_j}^{\pi}(s_t, a_t) = \mathbb{E}\Big[\sum_{k=t}^{\infty} \gamma^{k-t} \mathbb{1}_{\{s_k=s_i, a_k=a_j\}} \,\big|\, \pi, \mathcal{P}, s_t, a_t\Big], \qquad (8)$$

where $\mathbb{1}_{\{s_k=s_i, a_k=a_j\}}$ is the indicator function returning 1 if $s_k = s_i$ and $a_k = a_j$, and 0 otherwise.

Prior work considered the SR relative to either an $\epsilon$-greedy exploration policy [43] or a random policy [38, 44, 61]. Instead, we consider a different "successor policy" for every state-action pair $(s_i, a_j)$, and define the optimal successor policy as the one maximizing the respective S-function, i.e.,

$$\rho_{s_i a_j}^* := \arg\max_{\rho} \, S_{s_i a_j}^{\rho}(s, a). \qquad (9)$$

The switch from $\pi$ to $\rho$ is intentional: to maximize the S-function means to maximize the occurrence of $(s_i, a_j)$ under $\gamma$ discounting, which can been seen as visiting $(s_i, a_j)$ as fast as possible from any other state.[6] Thus, Eq. (9) is a suitable goal-conditioned policy $\rho^*$ discussed in Algorithm 1. Similar to the Q-function, we can learn an approximation of the S-functions using Q-Learning, i.e.,

$$\widehat{S}_{s_i a_j}(s_t, a_t) \leftarrow (1 - \alpha_t)\widehat{S}_{s_i a_j}(s_t, a_t) + \alpha_t(\mathbb{1}_{\{s_t=s_i, a_t=a_j\}} + \gamma\max_{a} \widehat{S}_{s_i a_j}(s_{t+1}, a)). \qquad (10)$$

Because the initial estimates of $\widehat{S}_{s_i a_j}$ can be inaccurate, in practice we let $\rho$ to be $\epsilon$-greedy over $\widehat{S}_{s^G a^G}$. That is, at every time step, the agent selects the action $a = \arg\max_a \widehat{S}_{s^G a^G}(s_t, a)$ with probability $1-\epsilon$, or a random action otherwise. As discussed in the previous section, any $\epsilon$-greedy policy with $\epsilon > 0$ is sufficient to meet the criteria of Theorem 1.

## 3.2 Directed Exploration via Successor-Functions: A Summary

Our exploration strategy can be applied to any off-policy algorithm where the temporal-difference update of Eq. (2) and (10) is guaranteed to convergence under the GLIE assumption (pseudocode for Q-Learning in Appendix A.5). Most importantly, our exploration does not suffer from the limitations of optimism discussed in Section 2.2 — *the agent will eventually explore all state-action pairs even when rewards are partially observable, because exploration is not influenced by the Q-function (and, thus, by rewards), but fully driven by the S-function.* Consider the example in Figure 1b again, and an optimistic agent (i.e., with high initial $\widehat{Q}$) that learns a reward model and uses its estimate when rewards are not observable. As discussed, classic exploration strategies will likely fail — their exploration is driven by $\widehat{Q}$ that can be highly inaccurate, either because of its optimistic estimate or because the reward model queried for Q-updates is inaccurate (and will stay so without proper exploration). This can easily lead to premature convergence to a suboptimal policy (to collect the left coin). On the contrary, if the agent follows our exploration strategy it will always push the button, discover the large coin, and eventually learn the optimal policy — it does not matter if $\widehat{Q}$ is initialized optimistically high and would (wrongly) believe that collecting the left coin is optimal, because the agent follows $\widehat{S}$. And even if $\widehat{S}$ is initialized optimistically (or even randomly), the indicator reward $\mathbb{1}$ of Eq. (10) *is always observable*, thus the agent can always update $\widehat{S}$ at every step. As $\widehat{S}$ becomes more accurate, the agent will eventually explore optimally, i.e., maximizing visitation occurrences — it will be able to visit states more efficiently, to observe rewards more frequently and update its reward model properly, and thus to update $\widehat{Q}$ estimates with accurate rewards. To summarize, the explicitly decoupling of exploration ($\widehat{S}$) and exploitation ($\widehat{Q}$) together with the use of SR (whose reward is always observable) is the key of our algorithm. Experiments in the next section empirically validate both the failure of classic exploration strategies and the success of ours. In Appendix B.4 we further report a deeper analysis on a benchmark similar to the example of Figure 1.

**Related Work.** In reward-free RL, Jin et al. [26] proposed an algorithm where the agent explores by maximizing SR rewards. After collecting a sufficient amount of data, the agent no longer interacts with the environment and uses the data to learn policies maximizing different reward functions. In model-based RL, the agent of Hu et al. [23] alternates between exploration and exploitation at every training episode. During exploration, the agent follows a goal-conditioned policy randomly sampled

---

[6]In deterministic MDPs, maximizing discounted occurrences is equivalent to minimizing the (expected) time-to-visit. However, this is not always true in stochastic MDPs, with discounted occurrences tending to ignore long time-to-visit outliers, making the agent more risk-seeing than expected time-to-visit. Nonetheless, in both cases, goal-conditioned policies have bounded goal-relative diameter as long as the MDP is communicating.

from a replay buffer with the goal of minimizing the number of actions needed to reach a goal. Finally, in sparse-reward RL, Parisi et al. [54] learn two value functions — one using extrinsic sparse rewards, one using dense visitation-count rewards — and combine them to derive a UCB-like exploration policy. While the use of SR of Jin et al. [26], the goal-conditioned policies of Hu et al. [23], and the interaction of two separate value functions of Parisi et al. [54] are related to our work, there are significant differences to what we presented in this paper. First, our directed exploration does not strictly separate exploration and exploitation into two phases [26] or between episodes [23], but rather interleaves them step-by-step using either the Q-function or the S-functions. However, the two value functions are never combined together [54], as the agent follows either one or the other according to the coefficient $\beta_t$. Second, our goal-conditioned policy is chosen according to the least-visited state-action pair, rather than randomly sampled from a set [26] or from a replay buffer [23]. Third, none of them have considered the setting of Mon-MDPs and partially observable rewards.

# 4 Experiments

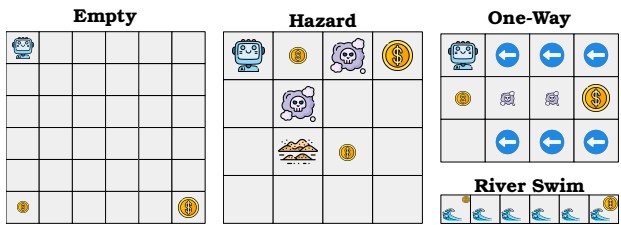

Figure 4: **Environments.** The goal is to collect the large coin ($r_t^E = 1$) instead of small "distracting" coins ($r_t^E = 0.1$). In Hazard, the agent must avoid quicksand (it prevents any movement) and toxic clouds ($r_t^E = -10$). In One-Way, the agent must walk over toxic clouds ($r_t^E = -0.1$) to get the large coin. In River Swim, the stochastic transition pushes the agent to the left. More details in Appendix A.2.

We validate our exploration strategy on tabular MDPs (Figure 4) characterized by different challenges, e.g., sparse rewards, distracting rewards, stochastic transitions. For each MDP, we propose the following Mon-MDP versions of increasing difficulty. The first has a simple **random monitor**, where positive and negative rewards are unobservable ($\hat{r}_t^E = \bot$) with probability $50\%$ and observable otherwise ($\hat{r}_t^E = r_t^E$), while zero-rewards are always observable. In the second, the agent can **ask to observe** the current reward at a cost ($r_t^M = -0.2$). In the third, the agent can turn monitoring ON/OFF by pushing a

**button** in the top-left cell, and if $s_t^M = \text{ON}$ the agent pays a cost ($r_t^M = -0.2$) and observes $\hat{r}_t^E = r_t^E$. In the fourth, there are **many experts** the agent can ask for rewards from at a cost ($r_t^M = -0.2$), but only a random one is available at the time. In the fifth and last, the agent has to **level up** the monitor state by doing a correct sequence of costly ($r_t^M = -0.2$) monitor actions to observe rewards (a wrong action resets the level). In all Mon-MDPs, the optimal policy does not need monitoring ($r_t^M = -0.2$ to observe $\hat{r}_t^E = r_t^E$). However, prematurely doing so would prevent observing rewards and, thus, learning $Q^*$. More experiments on more environments are presented in Appendix B.

**Baselines.** We evaluate Q-Learning with the following exploration policies (details in Appendix A.5): (1) **ours**; (2) greedy with **optimistic** initialization; (3) **naive** $\epsilon$-greedy; (4) $\epsilon$-greedy with count-based **intrinsic reward** [9]; (5) $\epsilon$-greedy with **UCB** bonus [4]; (6) $\epsilon$-greedy with **Q-Counts** [54]. Note that if rewards and transitions are deterministic and fully observable, then (2) is very efficient [17, 75, 77], thus serves as best-case scenario baseline. For all algorithms, $\gamma = 0.99$ and $\epsilon_t$ starts at 1 and linearly decays to 0. Because in Mon-MDPs environment rewards are always unobservable for some monitor state-action pairs (e.g., when the monitor is OFF), all algorithms learn a reward model to replace $\hat{r}_t^E = \bot$, initialized to random values in $[-0.1, 0.1]$ and updated when $\hat{r}_t^E \neq \bot$.[7]

**Why Is This Hard?** Q-function updates rely on the reward model, but the model is inaccurate at the beginning. To learn the reward model and produce accurate updates the agent must perform suboptimal actions and observe rewards. This results in a vicious cycle in exploration strategies that rely on the Q-function: the agent must explore to learn the reward model, but the Q-function will mislead the agent and provide unreliable exploration (especially if optimistic).

**Results.** For each baseline, we test the *greedy policy* at regular intervals (full details in Appendix A). Figure 5 shows that **Our** exploration outperforms all the baselines, being the only one converging to the optimal policy in the vast majority of the seeds for all environments and monitors. When

---

[7] Because for every environment reward there is a monitor state-action pair for which the reward is observable (i.e., $s_t^M = \text{ON}$), Q-Learning is guaranteed to converge given infinite exploration [55].

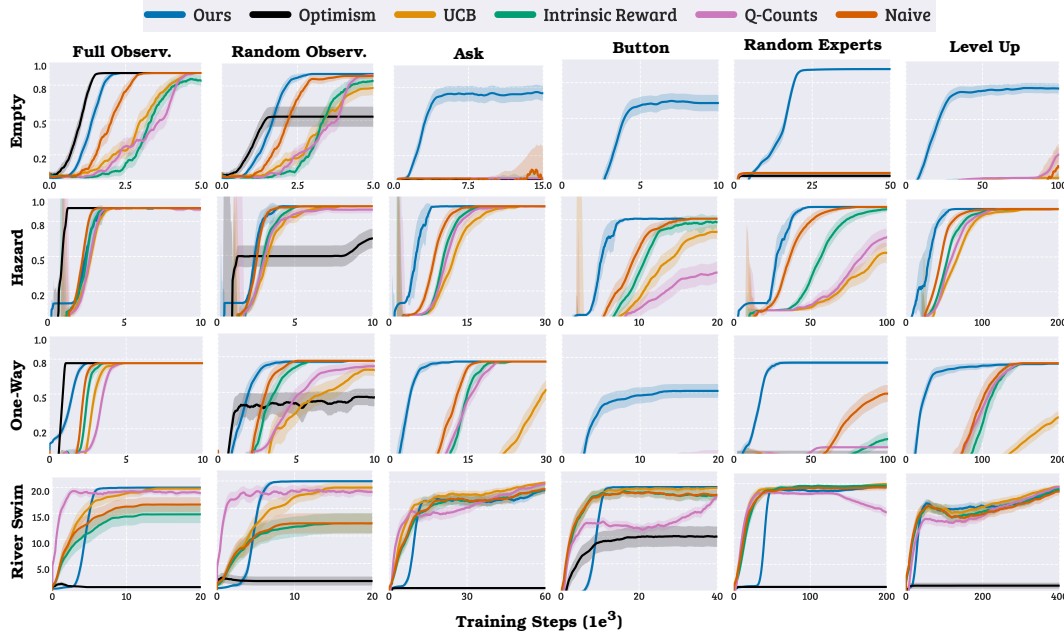

Figure 5: **Episode return** $\sum(r_t^{\mathrm{E}} + r_t^{\mathrm{M}})$ **of greedy policies averaged over 100 seeds (shades denote 95% confidence interval).** Our exploration clearly outperforms all baselines, as it is the only one learning in all Mon-MDPs. Indeed, while all baselines learn relatively quickly when rewards are fully observable (first column), their performance drastically decreases with rewards partial observability.

the environment is deterministic and rewards are always observable (first column), pure **Optimism** is optimal (as expected). However, if the environment is stochastic (River Swim) or rewards are unobservable, **Optimism** fails. Even just random observability of the reward (second column) is enough to cause convergence to suboptimal policies — without observing rewards the agent must rely on its model estimates, but these are initially wrong and mislead $\widehat{Q}$ updates, preventing exploration. Even though mitigated by the $\epsilon$-randomness, exploration with **Q-Counts**, **UCB**, **Intrinsic Reward**, and **Naive** is suboptimal as well, as these baselines learn suboptimal policies in most of the seeds, especially in harder Mon-MDPs. On the contrary, **Our** exploration is only slightly affected by the rewards unobservability because it relies on $\widehat{S}$ rather than $\widehat{Q}$ — the agent visits all state-action pairs as uniformly as possible, observes rewards frequently, and ultimately learns accurate $\widehat{Q}$ that make the policy optimal. Figure 6 strengthens these results, showing that indeed **Our** exploration observes significantly more rewards than the baselines in all Mon-MDPs — because we decouple exploration and exploitation, and exploration does not depend on $\widehat{Q}$, **Our** agent does not avoid suboptimal actions and discovers more rewards.[8] More plots and discussion in Appendix B. Source code at `https://github.com/AmiiThinks/mon_mdp_neurips24`.

## 5 Discussion

In this paper, we discussed the challenges of exploration when the agent cannot observe the outcome of its actions, and highlighted the limitations of existing approaches. While partial monitoring is a well-known and studied framework for bandits, prior work in MDPs with unobservable rewards is still limited. To fill this gap, we proposed a paradigm change in the exploration-exploitation trade-off and investigated its efficacy on Mon-MDPs, a general framework where the reward observability is governed by a separate process the agent interacts with. Rather than relying on optimism, intrinsic motivation, or confidence bounds, our approach explicitly decouples exploration and exploitation through a separate goal-conditioned policy that is fully in charge of exploration. We proved the convergence of this paradigm under some assumptions, presented a suitable goal-conditioned policy based on the successor representation, and validated it on a collection of MDPs and Mon-MDPs.

---

[8]Note that **Our** agent keeps exploring and observing rewards (Figure 6) because $\beta_t$ did not reach the exploit threshold $\bar{\beta}$ (but still decays quickly, see Appendix B.3), even if its greedy policy is already optimal (Figure 5).

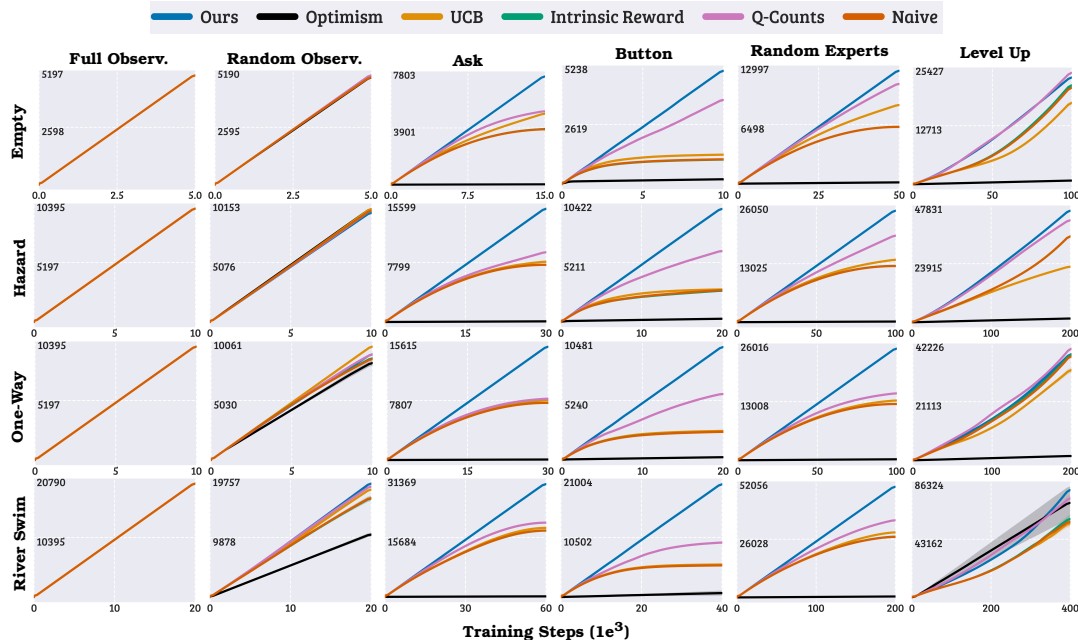

Figure 6: **Number of rewards observed ($\hat{r}_t^E \neq \perp$) during training by the exploration policies.** In Mon-MDPs, only **Ours** observes enough rewards because it does not avoid suboptimal actions.

**Advantages.** Directed exploration via the S-functions benefits from implicitly estimating the dynamics of the environment and being independent of the reward observability. First, the focus on the environment (states, actions, and dynamics) rather than on rewards or optimism (e.g., optimistic value functions) disentangles efficient exploration from imperfect or absent feedback — not observing the rewards may compromise exploratory strategies that rely on them. Second, the goal-conditioned policy learns to explore the environment regardless of the reward, i.e., of a task. This has potential application in transfer RL, in the same spirit of prior use of the successor representation [6, 19, 38, 61]. Third, the goal-conditioned policy can guide the agent to any desired goal state-action pair, and by choosing the least-visited pair as the goal the agent implicitly explores as uniformly as possible.

**Limitations and Future Work.** First, we considered tabular MDPs, thus we plan to follow up on continuous MDPs. This will require extending the S-function and the visitation count to continuous spaces, e.g., by following prior work on universal value function approximators [64], successor features [43, 61], and pseudocounts [42, 46, 78]. For example, the finite set of S-functions $\widehat{S}_{s_i a_j}(s_t, a_t)$ could be replaced by a neural network $\widehat{S}(s_t, a_t, s_i, a_j)$. For discrete actions, the network would take the (current state, goal state) pair and output the value for all (action, goal action) pairs, similarly to how deep Q-networks [48] work. Extending visitation counts to continuous spaces is more challenging, because Algorithm 1 relies on the min operator to select the goal. To make it tractable, one solution would be to store samples into a replay buffer and prioritize sampling according to pseudocounts, similarly to what prioritized experience replay does with TD errors [65].

Second, our exploration policy requires some stochasticity to explore as we learn the S-functions. In our experiments, we used $\epsilon$-greedy noise for a fair comparison against the baselines, but there may be better alternatives (e.g., "balanced wandering" [27]) that could improve our exploration.

Third, we assumed that the MDP has a bounded goal-relative diameter but this may not be true, e.g., in weakly-communicating MDPs [18, 59]. Thus, we will devote future work developing algorithms for less-constrained MDPs, following recent advances in convergence of reward-free algorithms [12].

Finally, we focused on model-free RL (i.e., Q-Learning) but we believe that our exploration strategy can benefit from model-based approaches. For example, we could use algorithms like UCRL [3, 24] *to learn the S-functions* (not the Q-function), whose indicator reward is always observable. And, by learning the model of the monitor, the agent could plan *what rewards to observe* (e.g., the least-observed) rather than what state-action pair to visit. Furthermore, our strategy could be combined with model-based methods where exploration is driven by uncertainty [45, 74] rather than visits. That would allow the agent to plan visits to states where rewards are more likely to be observed.

## Acknowledgments and Disclosure of Funding

This research was supported by grants from the Alberta Machine Intelligence Institute (Amii); a Canada CIFAR AI Chair, Amii; Digital Research Alliance of Canada; and NSERC.

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

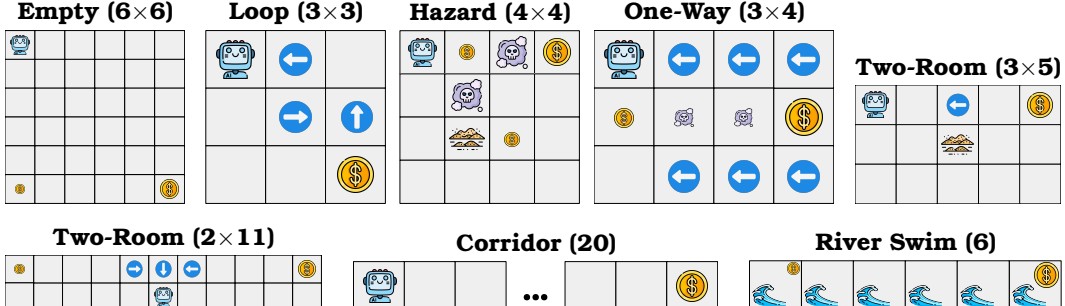

Figure 7: **Full set of the tabular environments used for our experiments.** Loop (3×3), Two-Room (3×5), Two-Room (2×11), and Corridor (20) were not included in Section 4 due to page limits.

## A  Experiments Details

### A.1  Source Code and Compute Details

Source code at `https://github.com/AmiiThinks/mon_mdp_neurips24`.
We ran our experiments on a SLURM-based cluster, using 32 Intel E5-2683 v4 Broadwell @ 2.1GHz CPUs. One single run took between 3 to 45 minutes on a single core, depending on the size of the environment and the number of training steps. Runs were parallelized whenever possible.

### A.2  Environments

In all environments except River Swim, the environment actions are $a^{\mathrm{E}} \in \{\mathrm{LEFT}, \mathrm{RIGHT}, \mathrm{UP}, \mathrm{DOWN}, \mathrm{STAY}\}$, and the agent starts in the position pictured in Figure 7. The environment reward depends on the current state and action as follows.

- Coin cells: STAY gives a positive reward ($r_t^{\mathrm{E}} = 0.1$ for small coins, $r_t^{\mathrm{E}} = 1$ for large ones) and end the episode.
- Toxic cloud cells: any action gives a negative reward ($r_t^{\mathrm{E}} = -0.1$ for small coins, $r_t^{\mathrm{E}} = -10$ for large ones).
- Any other cell: $r_t^{\mathrm{E}} = 0$.

The agent can move away from its current cell or stay, but some cells have special dynamics.

- In quicksand cells, any action can fail with 90% probability. These cells can easily prevent exploration, because the agent can get stuck and waste time.
- In arrowed cells, only the corresponding action succeeds. For example, if the agent is in a cell pointing to the left, the only action to move away from it is LEFT. In practice, they force the agent to take a specific path to reach the goal (e.g., in One-Way), or divide the environment in rooms (e.g., in Two-Rooms).

In all environments, the goal is to follow the shortest path to the large coin and STAY there. Below we discuss the characteristics and challenges of the first seven environments. Episodes end when the agent collects a coin or after a step limit (in brackets after the environment name and its grid size).

- **Empty (6×6; 50 steps).** This is a classic benchmark for exploration in RL with sparse rewards. There are no intermediate rewards between the initial position (top-left) and the large reward (bottom-right). Furthermore, the small coin in the bottom-left can "distract" the agent and cause convergence to a suboptimal policy.
- **Loop (3×3; 50 steps).** The large coin can be reached easily, but if the agent wants to visit the top-right cell is forced to follow the arrowed "loop". Algorithms that explore inefficiently or for too long can end up visiting those states too often.
- **Hazard (4×4; 50).** Because of the toxic clouds, the optimal path is to walk following the edges of the grid. Along the way, however, the agent may find the a small coin and converge to a suboptimal policie, or get stuck in the quicksand cell.
- **One-Way (3×4; 50 steps).** The only way to reach the large coin is to walk past small toxic clouds. Because of their negative reward, shallow exploration strategies can prematurely decide not to

explore further and never discover the large coin. The presence of an easy-to-reach small coin can also cause convergence to a suboptimal policy.

- **Corridor (20; 200 steps).** This is a flat variant of the environment presented by Klissarov and Machado [30]. Because of the length of the corridor, the agent requires persistent deep exploration from the start (leftmost cell) to the large coin (rightmost cell) to learn the optimal policy. Naive dithering exploration strategies like $\epsilon$-greedy are known to be extremely inefficient for this.
- **Two-Room (2×11; 200 steps)**. The arrowed cells divide the grid into two rooms, one with a small coin and one with a large coin. If exploration relies too much on randomness or value estimates, the agent may converge to the suboptimal policy collecting the small coin.
- **Two-Room (3×5; 50 steps)**. The arrowed cell divides the grid into two rooms, and moving from one room to another may be hard if the agent gets stuck in the quicksand cell.

River Swim is different from the other seven environments because the agent can only move `LEFT` and `RIGHT`, and there are no terminal states. This environment was originally presented by Strehl and Littman [72] and later became a common benchmark for exploration in RL, although often with different transition and reward functions. Here, we implement the version in Figure 8. There is a small positive reward for doing `LEFT` in the leftmost cell (acting as local optima) and a large one for doing `RIGHT` in the rightmost cell. The transition is stochastic and `RIGHT` has a high chance of failing in every cell, either not moving the agent or pushing it back to the left. The initial position is either state 1 or 2. Although this is an infinite-horizon MDP, i.e., no state is terminal, we reset episodes after 200 steps. The optimal policy always does `RIGHT`.

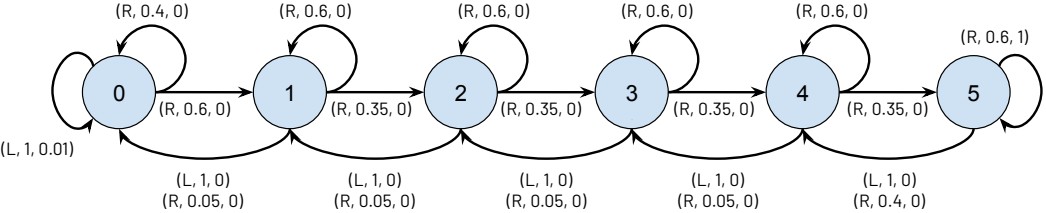

Figure 8: **River Swim reward and transition functions.** Each tuple represents (action, probability, reward) and arrows specify the corresponding transition. For example, doing `RIGHT` in state 0 has 60% chance of moving the agent to 1 and 40% chance of keeping it in 0. In both cases, the reward is 0. In state 5 the transition probability is the same, but if `RIGHT` succeeds the reward is 1.

### A.3 Monitored-MDPs

- **Random Monitor.** There is no monitor state, action, nor reward. Positive (coin) and negative (toxic cloud) environment rewards are unobservable with probability 50% ($\hat{r}_t^\text{E} = \perp$) and fully observable otherwise ($\hat{r}_t^\text{E} = r_t^\text{E}$). Zero-rewards in other cells are always observable.
- **Ask Monitor.** The monitor is always `OFF`. The agent can observe the current environment reward with an explicit monitor action at a cost.

$$\mathcal{S}^\text{M} \coloneqq \{\texttt{OFF}\} \qquad \mathcal{A}^\text{M} \coloneqq \{\texttt{ASK, NO-OP}\} \qquad s_{t+1}^\text{M} = \texttt{OFF}, \, t \geq 0$$

$$\hat{r}_t^\text{E} = \begin{cases} r_t^\text{E} & \text{if } a_t^\text{M} = \texttt{ASK} \\ \perp & \text{otherwise} \end{cases} \qquad r_t^\text{M} = \begin{cases} -0.2 & \text{if } a_t^\text{M} = \texttt{ASK} \\ 0 & \text{otherwise} \end{cases}$$

- **Button Monitor.** The monitor is turned `ON`/`OFF` by pushing a button in the starting cell (except for River Swim, where the button is in state 0). The agent can do that with $a^\text{E} = \texttt{LEFT}$, i.e., the agent has no dedicated monitor action. Having the monitor `ON` is costly, and leaving it `ON` when collecting a coin results in an extra cost. Because the initial monitor state is random, the optimal policy always turns it `OFF` before collecting the large coin.

$$\mathcal{S}^\text{M} \coloneqq \{\texttt{ON, OFF}\} \qquad \mathcal{A}^\text{M} \coloneqq \{\texttt{NO-OP}\} \qquad s_1^\text{M} = \text{random uniform in } \mathcal{S}^\text{M}$$

$$s_{t+1}^\text{M} = \begin{cases} \texttt{ON} & \text{if } s_t^\text{M} = \texttt{OFF} \text{ and } s_t^\text{E} = \texttt{BUTTON-CELL} \text{ and } a_t^\text{E} = \texttt{LEFT} \\ \texttt{OFF} & \text{if } s_t^\text{M} = \texttt{ON} \text{ and } s_t^\text{E} = \texttt{BUTTON-CELL} \text{ and } a_t^\text{E} = \texttt{LEFT} \\ s_t^\text{M} & \text{otherwise} \end{cases}$$

$$\hat{r}_t^{\mathrm{E}} = \begin{cases} r_t^{\mathrm{E}} & \text{if } s_t^{\mathrm{M}} = \mathtt{ON} \\ \bot & \text{otherwise} \end{cases} \qquad r_t^{\mathrm{M}} = \begin{cases} -0.2 & \text{if } s_t^{\mathrm{M}} = \mathtt{ON} \\ -2 & \text{if } s_t^{\mathrm{M}} = \mathtt{ON} \text{ and } s_t^{\mathrm{E}} \text{ is terminal} \\ 0 & \text{otherwise} \end{cases}$$

- **Random Experts Monitor.** At every step, the monitor state is random. If the agent's monitor action matches the state, it observes $\hat{r}_t^{\mathrm{E}} = r_t^{\mathrm{E}}$ but receives a negative monitor reward. Otherwise, it receives $\hat{r}_t^{\mathrm{E}} = \bot$ but receives a smaller positive monitor reward.

$$\mathcal{S}^{\mathrm{M}} \coloneqq \{1, \ldots, n\} \qquad \mathcal{A}^{\mathrm{M}} \coloneqq \{1, \ldots, n\} \qquad s_{t+1}^{\mathrm{M}} = \text{random uniform in } \mathcal{S}^{\mathrm{M}}, \ t \geq 0$$

$$\hat{r}_t^{\mathrm{E}} = \begin{cases} r_t^{\mathrm{E}} & \text{if } a_t^{\mathrm{M}} = s_t^{\mathrm{M}} \\ \bot & \text{otherwise} \end{cases} \qquad r_t^{\mathrm{M}} = \begin{cases} -0.2 & \text{if } a_t^{\mathrm{M}} = s_t^{\mathrm{M}} \\ 0.001 & \text{otherwise} \end{cases}$$

In our experiments, $n = 4$. An optimal policy collects the large coin while never matching monitor states and actions, in order to receive small positive monitor rewards along the way. However, prematurely deciding to be greedy with respect to monitor rewards will result in never observing environment rewards, thus not learning how to collect coins. We also note that this monitor has larger state and action spaces than the others, and a stochastic transition.

- **Level Up Monitor.** The monitor has $n$ states, each representing a "level", and $n + 1$ actions. The first $n$ actions are costly and needed to "level up" the state, while the last is $\mathtt{NO\text{-}OP}$. The initial level is random. To level up, the agent's action must match the state, e.g., if $s_t^{\mathrm{M}} = 1$ and $a_t^{\mathrm{M}} = 1$, then $s_{t+1}^{\mathrm{M}} = 2$. However, if the agent executes the wrong action, the level resets to $s_{t+1}^{\mathrm{M}} = 1$. Environment rewards will become visible only when the monitor level is $n$. Leveling up the monitor is costly, and only $a_t^{\mathrm{M}} = \mathtt{NO\text{-}OP}$ is cost-free.

$$\mathcal{S}^{\mathrm{M}} \coloneqq \{1, \ldots, n\} \qquad \mathcal{A}^{\mathrm{M}} \coloneqq \{1, \ldots, n, \mathtt{NO\text{-}OP}\}$$

$$s_{t+1}^{\mathrm{M}} = \begin{cases} \max(s_t^{\mathrm{M}} + 1, n) & \text{if } s_t^{\mathrm{M}} = a_t^{\mathrm{M}} \\ s_t^{\mathrm{M}} & \text{if } a_t^{\mathrm{M}} = \mathtt{NO\text{-}OP} \\ 1 & \text{otherwise} \end{cases}$$

$$\hat{r}_t^{\mathrm{E}} = \begin{cases} r_t^{\mathrm{E}} & \text{if } s_t^{\mathrm{M}} = n \\ \bot & \text{otherwise} \end{cases} \qquad r_t^{\mathrm{M}} = \begin{cases} 0 & \text{if } a_t^{\mathrm{M}} = \mathtt{NO\text{-}OP} \\ -0.2 & \text{otherwise} \end{cases}$$

The optimal policy always does $a_t^{\mathrm{M}} = \mathtt{NO\text{-}OP}$ to avoid negative monitor rewards. Yet, without leveling up the monitor, the agent cannot observe rewards and learn about environment rewards in the first place. Furthermore, the need to do a correct sequence of actions to observe rewards elicits deep exploration.

## A.4 Training Steps and Testing Episodes

All algorithms are trained over the same amount of steps, but this depends on the environment and the monitor as reported in Table 1. As the agent explores and learns, we test the greedy policies at regular intervals for a total of 1,000 testing points. For example, in River Swim with the Button Monitor the agent learns for 40,000 steps, therefore we test the greedy policy every 40 training steps. Every testing point is an evaluation of the greedy policy over 1 episode (for fully deterministic environments and monitors) or 100 (for environments and monitors with stochastic transition or initial state, i.e., Hazard, Two-Room (3×5), River Swim; Button, Random Experts, Level Up).

Table 1: **Number of training steps.** For each environment, we decided a default number of training steps. Then, we multiplied this number for a constant depending on the difficulty of the monitor. For example, agents are trained for 30,000 steps in the Corridor environment without any monitor, 60,000 with the Button Monitor, and 600,000 with the Level Up Monitor.

| Environment | Default Steps |
|---|---|
| Empty (6×6) | 5,000 |
| Loop (3×3) | 5,000 |
| Hazard (4×4) | 10,000 |
| One-Way (3×4) | 10,000 |
| Corridor (20) | 30,000 |
| Two-Room (2×11) | 5,000 |
| Two-Room (3×5) | 10,000 |
| River Swim (6) | 20,000 |

| Monitor | Multiplier |
|---|---|
| Full Obs. | ×1 |
| Random | ×1 |
| Ask | ×3 |
| Button | ×2 |
| Rnd. Experts | ×10 |
| Level Up | ×20 |

**Algorithm 2:** Q-Update With Reward Model for Mon-MDPs

**input :** $\{s_t^{\mathrm{E}}, a_t^{\mathrm{E}}, \hat{r}_t^{\mathrm{E}}, s_{t+1}^{\mathrm{E}}, s_t^{\mathrm{M}}, a_t^{\mathrm{M}}, r_t^{\mathrm{M}}, s_{t+1}^{\mathrm{M}}\}, \alpha_t, N^r, \widehat{R}, \widehat{Q}, \gamma$

1 **if** $\hat{r}_t^{\mathrm{E}} \neq \perp$ **then**

2     $N^r(s_t^{\mathrm{E}}, a_t^{\mathrm{E}}) \leftarrow N^r(s_t^{\mathrm{E}}, a_t^{\mathrm{E}}) + 1$

3     $\widehat{R}(s_t^{\mathrm{E}}, a_t^{\mathrm{E}}) \leftarrow \frac{(N^r(s_t^{\mathrm{E}}, a_t^{\mathrm{E}}) - 1)\widehat{R}(s_t^{\mathrm{E}}, a_t^{\mathrm{E}}) + \hat{r}_t^{\mathrm{E}}}{N^r(s_t^{\mathrm{E}}, a_t^{\mathrm{E}})}$

4 $s_t \leftarrow (s_t^{\mathrm{E}}, s_t^{\mathrm{M}})$

5 $a_t \leftarrow (a_t^{\mathrm{E}}, a_t^{\mathrm{M}})$

6 $r_t \leftarrow \widehat{R}(s_t^{\mathrm{E}}, a_t^{\mathrm{E}}) + r_t^{\mathrm{M}}$

7 **return** $\widehat{Q}(s_t, a_t) \leftarrow (1 - \alpha_t)\widehat{Q}(s_t, a_t) + \alpha_t(r_t + \gamma \max_a \widehat{Q}(s_{t+1}, a))$

### A.5 Baselines

All baselines evaluated in Section 4 use Q-Learning with reward model to compensate for unobservable rewards in Mon-MDPs ($\hat{r}_t^{\mathrm{E}} = \perp$), as outlined in Algorithm 2. Because there is always a monitor state-action pair for which all environment rewards are observable, Q-Learning with reward model will converge given enough exploration [55]. The reward model is a table $\widehat{R}(s^{\mathrm{E}}, r^{\mathrm{E}})$ initialized to random values in $[-0.1, 0.1]$ that keeps track of the running average of observed proxy reward — every time a reward is observed, a count $N^r(s^{\mathrm{E}}, s^{\mathrm{E}})$ is increased and the entry $\widehat{R}(s^{\mathrm{E}}, r^{\mathrm{E}})$ is updated.

The differences between the baselines lie in their exploration policies, as described below and outlined in Algorithms 3, 4 and 5.

- **Ours.** As described in Algorithm 1, with $\bar{\beta} = 0.01$. The goal-conditioned policy $\rho$ is $\epsilon$-greedy over $\widehat{S}_{s^{\mathrm{G}} a^{\mathrm{G}}}(s_t, a)$. The S-function is initialized optimistically to $\widehat{S}_{s_i a_j}(s, a) = 1 \; \forall(s, a, s_i, a_j)$.

- **Optimism.** Exploration is pure greedy, i.e., $a_t = \arg\max_a \widehat{Q}(s_t, a)$.

- **UCB.** Exploration is $\epsilon$-greedy over the UCB estimate $\widehat{Q}(s_t, a) + \sqrt{2 \log \sum_a N(s_t, a)/N(s_t, a)}$ [4].

- **Q-Counts.** This baseline learns a separate Q-function $\widehat{Q}_{\mathrm{c}}(s, a)$ using $N(s_t, a_t)$ as rewards, i.e.,

$$\widehat{Q}_{\mathrm{c}}(s_t, a_t) \leftarrow (1 - \alpha_t)\widehat{Q}_{\mathrm{c}}(s_t, a_t) + \alpha_t\left(N(s_t, a_t) + \gamma\min_a \widehat{Q}_{\mathrm{c}}(s_{t+1}, a)\right). \tag{11}$$

  This function is initialized to $\widehat{Q}_{\mathrm{c}}(s, a) = 0 \; \forall(s, a)$, and intuitively estimates and minimizes "cumulative visitation counts" [54]. The exploration policy then replaces $N(s, a)$ with $\widehat{Q}_{\mathrm{c}}(s, a)$ in the UCB estimate, favoring actions that result in low cumulative counts rather than low immediate counts. Formally, the exploration is $\epsilon$-greedy over $\widehat{Q}(s_t, a) + \sqrt{2 \log \sum_a \widehat{Q}_{\mathrm{C}}(s_t, a)/\widehat{Q}_{\mathrm{C}}(s_t, a)}$.

- **Intrinsic.** Exploration is $\epsilon$-greedy over $\widehat{Q}(s_t, a)$. The Q-function is trained with intrinsic reward $0.01/\sqrt{N(s_t, a_t)}$ [9].

- **Naive.** Exploration is $\epsilon$-greedy over $\widehat{Q}(s_t, a)$.

When two or more actions have the same value estimate — whether $\widehat{Q}(s_t, a)$, $\widehat{S}_{s^{\mathrm{G}} a^{\mathrm{G}}}(s_t, a)$, or $\widehat{Q}(s_t, a)$ with UCB bonuses) — ties are broken randomly.

Below are the learning hyperparameters.

- The schedule $\epsilon_t$ starts at 1 and linearly decays to 0 throughout all training steps. For all algorithms, this performed better than the following schedules: linear decay from 0.5 to 0, linear decay from 0.2 to 0, constant 0.2, constant 0.1.

- The schedule $\alpha_t$ depends on the environment: constant 0.5 for Hazard and Two-Room ($3 \times 5$) (because of the quicksand cell), linear decay from 0.5 to 0.05 in River Swim (because of the stochastic transition), and constant 1 otherwise. For the Random Experts Monitor we linearly decay the learning rate to 0.1 in all environments (0.05 in River Swim) because of the random monitor state.

- Discount factor $\gamma = 0.99$.

**Algorithm 3:** Q-Learning Baselines

**input:** $\alpha, N, \widehat{Q}$

1   $s_0 \leftarrow$ `environment.start()`
2   **for** $t = 0$ **to** $t_{\text{MAX}}$ **do**
3     $a_t = \texttt{explore}(s_t, \widehat{Q}, N)$
4     $N(s_t, a_t) \leftarrow N(s_t, a_t) + 1$
5     $r_t, s_{t+1} \leftarrow$ `environment.step`$(a_t)$
6     **if** $s_t$ is terminal **then**
7       $s_{t+1} \leftarrow$ `environment.start()`
8     `q_update`$(s_t, a_t, r_t, s_{t+1}, \alpha)$      `// Eq. (2)`

---

**Algorithm 4:** Our Q-Learning with S-functions

**input:** $\alpha, N, \widehat{Q}, \widehat{S}, \bar{\beta}$

1   $s_0 \leftarrow$ `environment.start()`
2   **for** $t = 0$ **to** $t_{\text{MAX}}$ **do**
3     $a_t = \texttt{explore}(s_t, \widehat{Q}, N, \widehat{S}, \bar{\beta})$
4     $N(s_t, a_t) \leftarrow N(s_t, a_t) + 1$
5     $r_t, s_{t+1} \leftarrow$ `environment.step`$(a_t)$
6     **if** $s_t$ is terminal **then**
7       $s_{t+1} \leftarrow$ `environment.start()`
8     `q_update`$(s_t, a_t, r_t, s_{t+1}, \alpha)$      `// Eq. (2)`
9     `s_update`$(s_t, a_t, s_{t+1}, \alpha)$      `// Eq. (10)`

---

**Algorithm 5:** Q-Learning with Q-Counts [54]

**input:** $\alpha, N, \widehat{Q}, \widehat{Q}_c$

1   $s_0 \leftarrow$ `environment.start()`
2   **for** $t = 0$ **to** $t_{\text{MAX}}$ **do**
3     $a_t = \texttt{explore}(s_t, \widehat{Q}, \widehat{Q}_c)$
4     $N(s_t, a_t) \leftarrow N(s_t, a_t) + 1$
5     $r_t, s_{t+1} \leftarrow$ `environment.step`$(a_t)$
6     **if** $s_t$ is terminal **then**
7       $s_{t+1} \leftarrow$ `environment.start()`
8     `q_update`$(s_t, a_t, r_t, s_{t+1}, \alpha)$      `// Eq. (2)`
9     `q_count_update`$(s_t, a_t, s_{t+1}, N, \alpha)$      `// Eq. (11)`

Note that the discount factor $\gamma$ and the learning rate $\alpha_t$ in Eq. (2), (10), and (11) can be different, e.g., a higher learning rate to update $\widehat{Q}$ and a smaller for $\widehat{S}$. In our experiment, we use the same $\gamma$ and $\alpha_t$ for both $\widehat{Q}$, $\widehat{S}$, and $\widehat{Q}_c$.

The Q-function is initialized optimistically to $\widehat{Q}(s, a) = 1 \,\forall(s, a)$ (first seven environments) and $\widehat{Q}(s, a) = 50 \,\forall(s, a)$ (River Swim) for all algorithms except **Ours**, where it is initialized pessimistically to $\widehat{Q}(s, a) = -10 \,\forall(s, a)$ (all environments). This performed better than optimistic initialization because it mitigates overestimation bias, that can be more prominent with optimism [49]. From our experiments, this is especially true in Mon-MDPs, where the reward model used to compensate for unobservable rewards can be inaccurate and lead to overly optimistic value estimates. We also tried the same pessimistic $\widehat{Q}$ initialization for **UCB**, **Q-Counts**, **Intrinsic**, and **Naive**, but their performance was significantly worse. Most of the time, in fact, the "distracting" small coins were causing premature convergence to suboptimal policies.

# B Additional Results

## B.1 Greedy Policy Return

Figure 9 is an extended version of Figure 5 with all eight environments (we showed only four in the main paper due to page limit). Results on Loop (3×3), Two-Room (3×5), Two-Room (2×11), and Corridor (20) are aligned with the remainder, showing that **Our** exploration strategy outperforms all the baselines. Note that only in River Swim, **Ours** did not *significantly* outperformed the baselines. Its highly stochastic transition, indeed, makes learning harder and is the reason why **Optimism** fails even in the MDP version (in spite of fully observable rewards). Nonetheless, **Ours** did not perform worse than any baseline in any version of River Swim (MDP and Mon-MDPs).

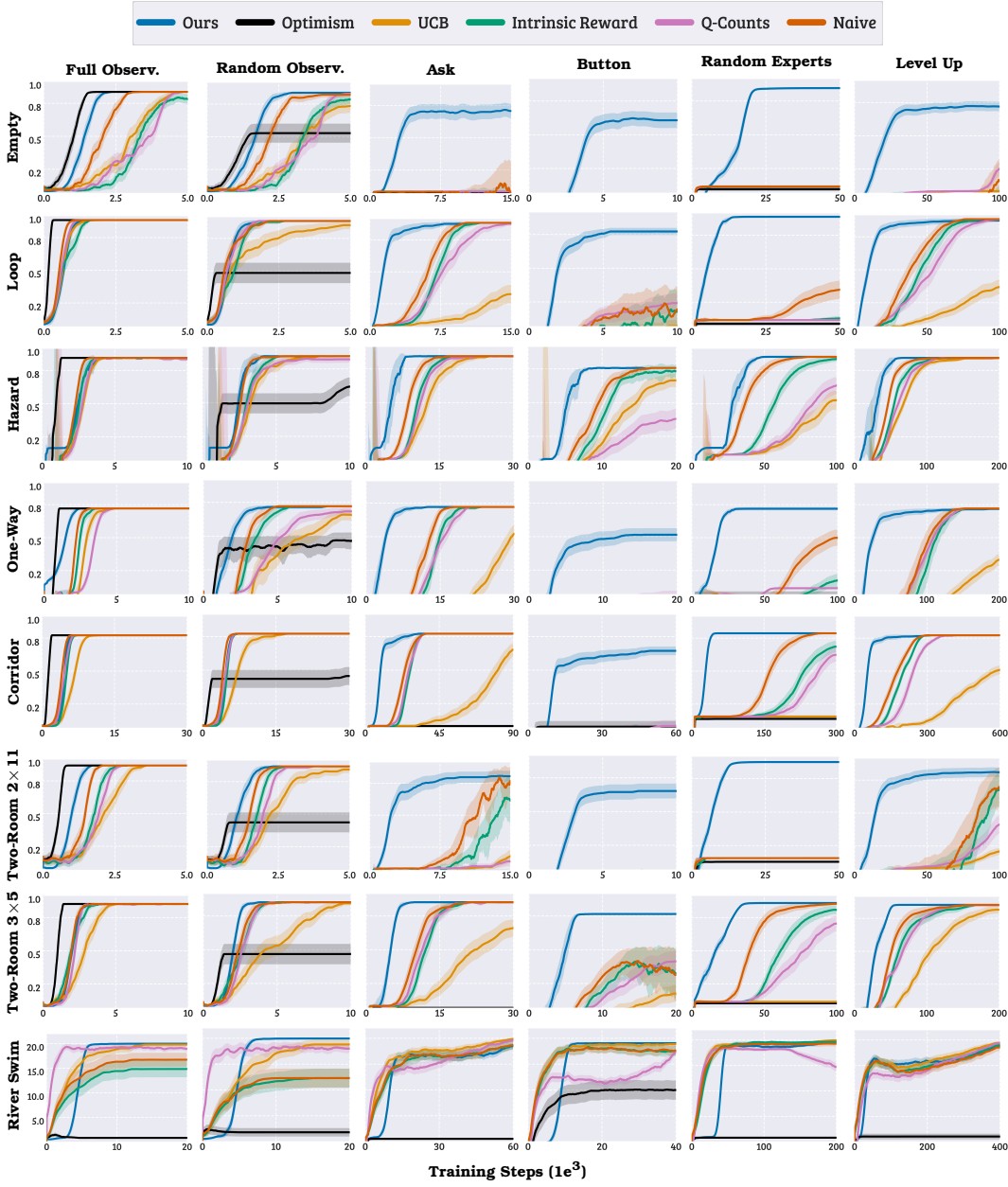

Figure 9: **Episode return** $\sum(r_t^{\mathbf{E}} + r_t^{\mathbf{M}})$ **of greedy policies averaged over 100 seeds (shades denote 95% confidence interval).**

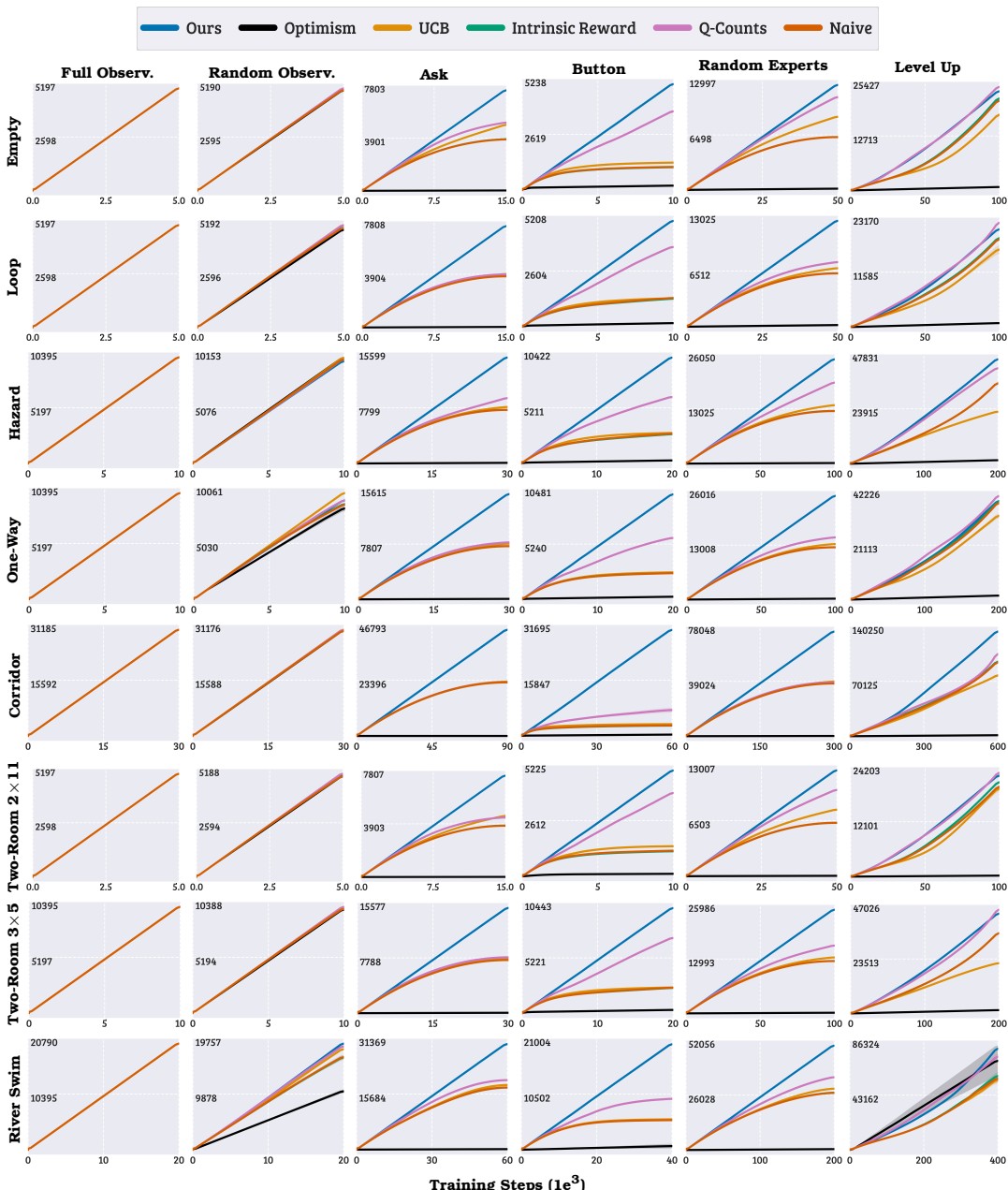

Figure 10: **Number of rewards observed ($\hat{r}_t^{\mathrm{E}} \neq \bot$) during training by the exploration policies averaged over 100 seeds (shades denote 95% confidence interval).** In Mon-MDPs, all baselines except **Ours** do not observe enough rewards because they do not perform suboptimal actions to explore. Thus, they often converge to suboptimal policies, as also shown in Figure 9. The ratio between training steps and number of observed rewards also hints that **Our** policy truly explores uniformly. For example, in **Ask** the agent can observe rewards only in half of the Mon-MDP state space (only when $s_t^{\mathrm{M}} = \mathrm{ON}$). Indeed, in the plots in the third column, the number of observed rewards is roughly half of the number of timesteps (e.g., in **Empty (Ask)** the agent is trained for 15,000 steps and **Our** agent ends up observing ∼7,500 rewards). The same $1/2$ ratio is consistent across all **Ask** plots. Similarly, in **Random Experts** the agent can observe rewards only in $1/4$ of the state space (there are four monitor states and only by matching the current state the agent can observe the reward), and indeed the agent ends up observing rewards ∼$1/4$ of the time (e.g., ∼12,500 out of 50,000 steps in **Empty (Random Experts)**). In Figure 13, we confirm exploration uniformity with heatmaps.

## B.2 Reward Discovery

Figure 10 shows how many rewards the agent observes ($\hat{r}_t^{\mathrm{E}} \neq \perp$) during training. When rewards are fully observable (first column) this is equal to the number of training steps. If reward observability is random and the agent cannot control it (second column), all the baselines still perform relatively well. However, when the agent has to *act* to observe rewards — and these actions are suboptimal, e.g. costly requests or pushing a button — results clearly show that **Optimism**, **UCB**, **Q-Counts**, **Intrinsic**, and **Naive** do not observe many rewards. Most of the time, in fact, they explore the environment only when rewards are observable, and therefore converge to suboptimal policies. As discussed throughout the paper, in fact, if the agent must perform suboptimal actions to gain information, classic approaches are likely to fail. **Our** exploration strategy, instead, actively explores all state-action pairs uniformly, including the ones where actions are suboptimal yet allow to observe rewards.

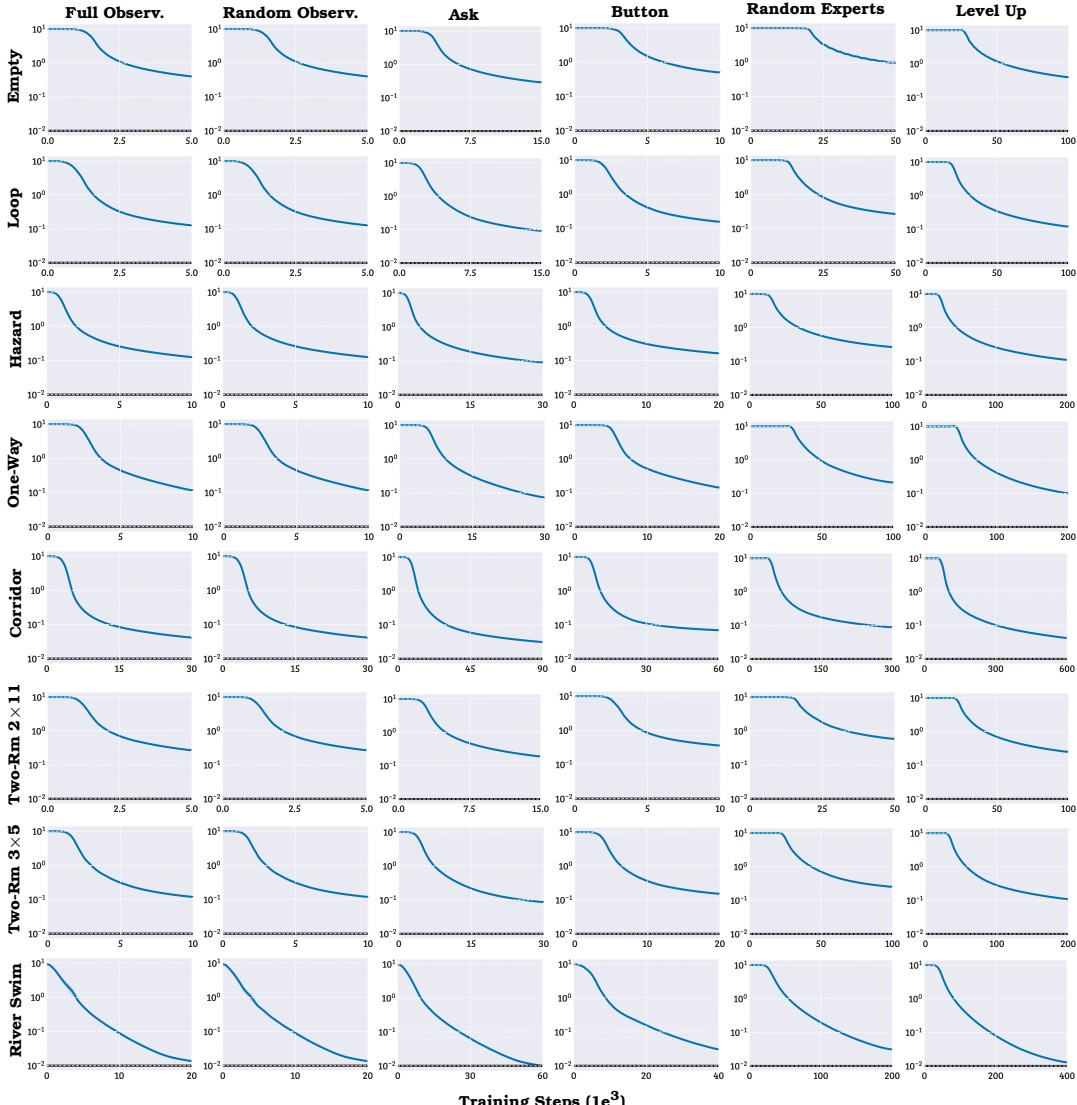

Figure 11: **Trend of the exploration-exploitation ratio $\beta_{\mathrm{t}}$ (in log-scale) during training by Our exploration policy averaged over 100 seeds (shades denoting 95% confidence are too small to be seen).** Until the agent has visited every state-action pair once, $\min N(s, a) = 0$, thus $\beta_t = \infty$. For the sake of clarity, we report it as $\beta_t = 10$ in this plot. Even if it did not reach the threshold $\bar{\beta} = 0.01$ (bottom line in black), the ratio clearly decreases over time, a further sign that the agent explores all state-action pairs uniformly (i.e., maximizing the occurrence of the least-visited pair). Note that in small environments (e.g., River Swim) $\beta_t$ decreases faster than in large environment (e.g., Empty).

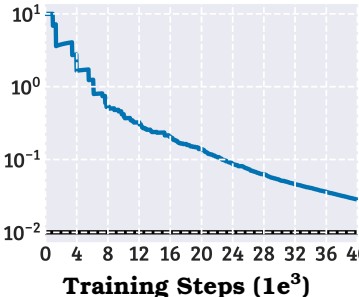

Figure 12: **Explore-exploit ratio** $\beta_t = \log t / \min N(s,a)$ (in log-scale) during a run on RiverSwim with Button Monitor. For the sake of clarity, we replaced $\beta_t = \infty$ (when $\min N(s,a) = 0$) with $\beta_t = 10$. For roughly the first 8,000 steps, $\beta_t$ trend is "irregular" — it goes down when the current state-action goal is visited, and then up until the new goal is visited. Later, these "up/down steps" become smaller and smaller and $\beta_t$ consistently decreases. This denotes that $\widehat{S}$ become more and more accurate, thus the policy $\rho$ can bring the agent to the goal (i.e., the least-visited state-action pair) faster.

## B.3 Goal-Conditioned Threshold $\beta_t$

Figure 11 shows the trend of the exploration-exploitation ratio $\beta_t$ (in log-scale) of **Our** exploration policy. The decay denotes that, indeed, the policy explores uniformly and efficiently, maximizing the occurrence of the least-visited state-action pair. The fact that $\beta_t$ did not reach the threshold $\bar{\beta}$ explains why in Figure 10 the agent keeps exploring and observing rewards, while the greedy policy in Figure 9 is already optimal. In Figure 12 we further show the trend of $\beta_t$ on one run on River Swim with the Button Monitor. After some time spent learning the S-functions $\widehat{S}$, the trend consistently decreases, denoting that the agent has learned to quickly visit any state-action pair.

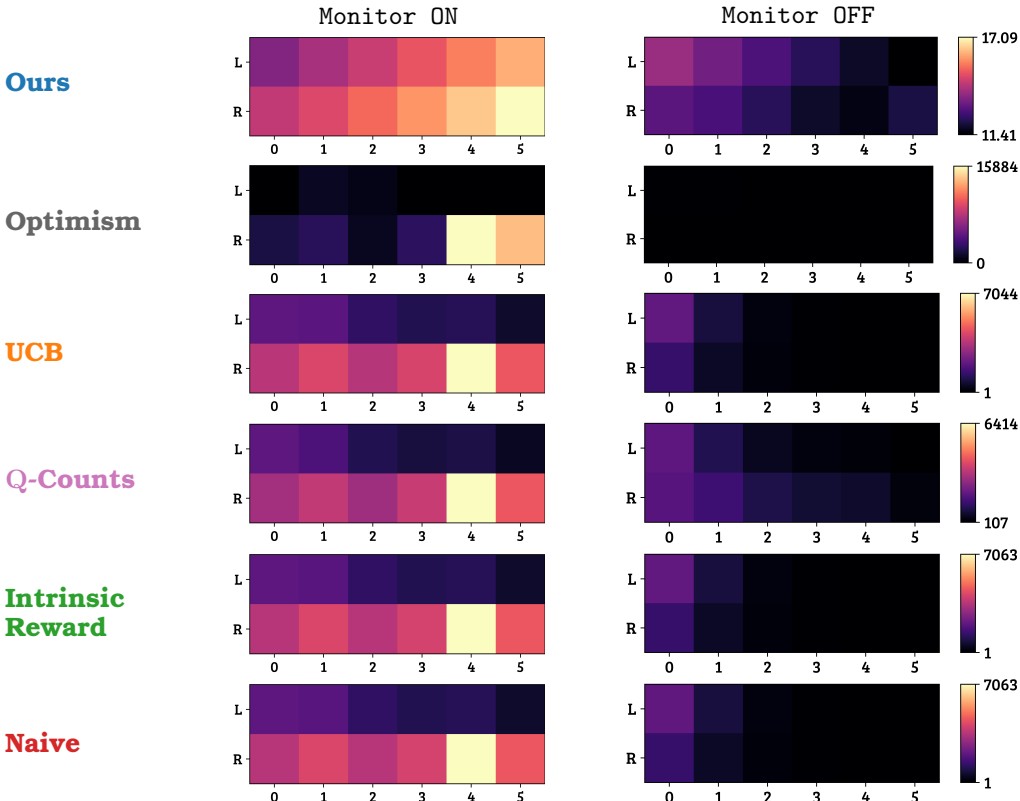

Figure 13: **State-action visitation counts** $\mathbf{N(s,a)}$ after one run (40,000 steps) on River Swim with the Button Monitor. Only **Our** agent visits the state-action space as uniformly as possible, as shown by the smooth heatmap and its range ($374-3,425$). On the contrary, all the other baselines explore poorly (less smooth heatmaps and much larger range). Indeed, they do not visit most environment state-action pairs when the monitor is ON (some have never been visited and have count 0). Having the monitor ON, in fact, is suboptimal ($r_t^M = -0.2$), yet needed to observe rewards. This further explains why they observe much fewer rewards in Figure 10.

## B.4  River Swim With Button Monitor: A Deeper Analysis

Here, we investigate more in detail the results for River Swim with Button Monitor, a benchmark similar to the example in Figure 1 discussed throughout the paper. States are enumerated from 0 to 5 as in Figure 8, the agent starts either in state 1 or 2, the button is located in state 0 and can be turned ON/OFF by doing LEFT in state 0. Figure 13 shows the visitation count and Figure 14 the Q-function for all state-action pairs. **Optimism** barely explores with monitoring ON. This is expected, because observing rewards is costly (negative monitor reward), and pure optimistic exploration does not select suboptimal actions (but this is the only way to discover higher rewards). **UCB**, **Q-counts**, **Intrinsic**, and **Naive** mitigate this problem thanks to $\epsilon$-randomness, but still do not explore properly and, ultimately, do not learn accurate Q-functions. On the contrary, **Our** policy explores the whole state-action space as uniformly as possible[9], observes rewards frequently (because it visits states with monitoring ON), and learns a close approximation of $Q^*$ that results in the optimal greedy policy.

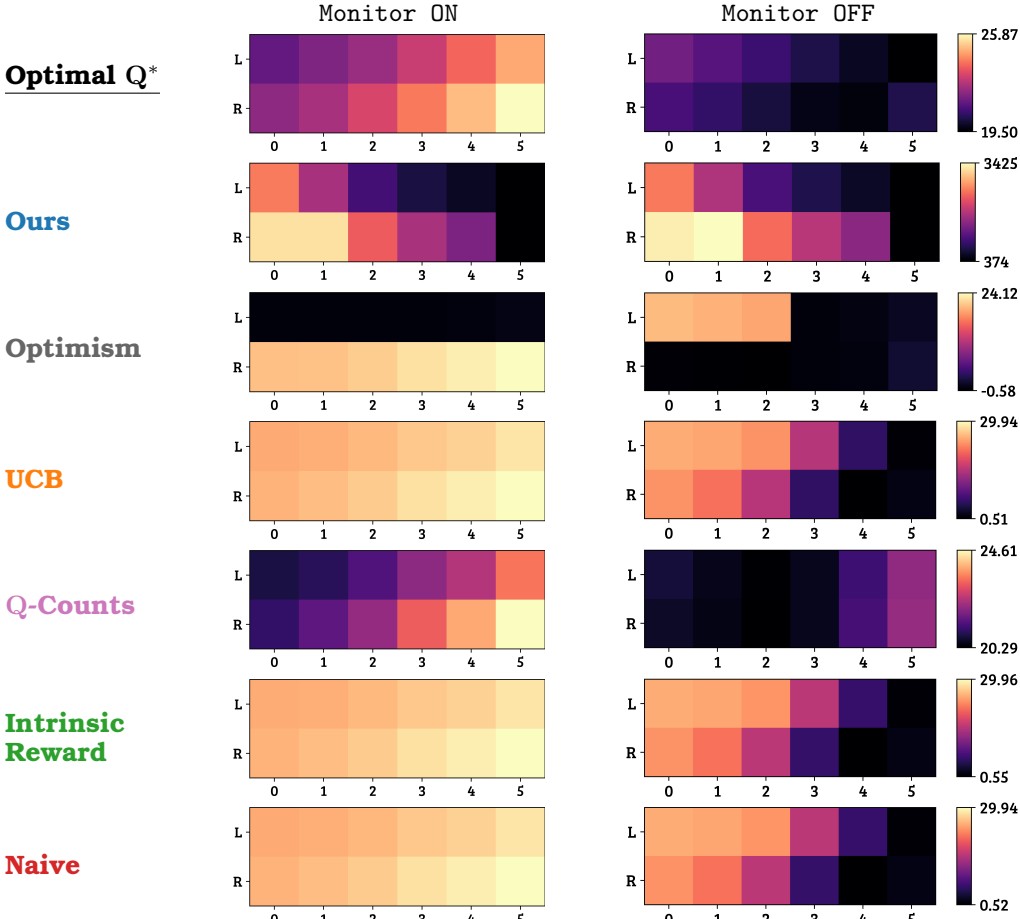

Figure 14: **Q-function approximators** $\widehat{Q}(s, a)$ after one run (40,000 steps) on River Swim with the Button Monitor. Only **Our** agent learns a close approximation of the true optimal Q-function (on the top) that produces the optimal behavior — $\widehat{Q}(5, \text{OFF}, \text{RIGHT})$ and $\widehat{Q}(0, \text{ON}, \text{LEFT})$ have the highest Q-value for monitoring OFF/ON, respectively, and the value of other state-action pairs smoothly decreases as the agent is further away from those states. Doing RIGHT in state 5, in fact, gives $r_t^{\text{E}} = 1$, but the optimal policy turns monitoring OFF first by doing LEFT in state 0 (to stop $r_t^{\text{M}} = -0.2$). On the contrary, **Optimism** learns a completely wrong Q-function that cannot turn monitoring OFF. This is not surprising given the poor visitation count of Figure 13. **UCB**, **Intrinsic**, and **Naive** learn smoother Q-function, but end up overestimating the Q-values due to the optimistic initialization and overestimation bias (see the range of their Q-functions).

---

[9]Note that the agent starts in state 1 or 2 and the transition is likely to push the agent to the left, thus states on the left are naturally visited more often.

