# OpenReview forum: "Beyond Optimism: Exploration With Partially Observable Rewards"
_NeurIPS.cc/2024/Conference — NeurIPS 2024 poster_

### Official Review · Reviewer_Y2Vd · 2024-06-25

**Soundness:** 3
**Presentation:** 3
**Contribution:** 2
**Rating:** 6
**Confidence:** 3

**Summary:**

The paper studies the setting of finite state action MDPs with partially observable rewards. To formalize the framework they introduce Monitored MDPs. They introduce the algorithm that separates exploration and exploitation and prove that for a class of MDPs with finite goal-oriented diameter their algorithm is greedy in the limit with probability 1. For exploration they use Successor Function that maximizes the cumulative discounted occurrences of the targeted state-action pair. In their experiments they show that their algorithm successfully solves all the tried tasks, while the baselines in most cases don't, or solve with with considerably more training steps.

**Strengths:**

- The algorithm is simple and and intuitive
- The paper is generally well-written and I didn't have issues with understanding.
- In the experiment section the algorithm is tested on several environments with different monitors, and they show significant improvements in terms of performance compared to the baselines.

**Weaknesses:**

- When you introduce the "unobserved reward" it would be great to emphasize that that this is not a reward 0, but really the agent knows it hasn't observed it, i.e. he gets the reward $\perp$.
- For me it was not clear what is the agent state (i.e. what agents sees) in the setting of Monitored MDPs. Does he have acess to both the state of the environment and the monitor or just the environment? As far as I could understand it has to both and I would emhphasise that in the section where you introduce the Monitored MDPs.
- The statement that starts in line 174: "As the agent explores, if every state-action pair is visited infinitely often, $\log(t)$ will grow at a slower rate than $N_t(s, a)$, ..." is wrong. It might apply to your algorithm but as you state here is sounds like it holds in general, but that is not true. Consider the case when you visit one state only at the rate $\log(\log(t))$, you still visit it infinitely often but the visition rate is slower than $\log(t)$.
- To me corollary 1 is not clear why it holds. I understand the theorem 1, where you say that with your algorithm you will visit every state action pair infinitely often in the limit and that your policy is greedy in the limit, but I don't see why $\hat{Q}$ converges to $Q^*$, how do you update your $\hat{Q}$ from the data? Does the state in $Q$ function depend only on environment state or does it depends also on monitor state? (same for action)
- In line 240 you claim that with the policy that comes from successor-function you will visit the targeted state as fast as possible. I think that is not true. With this policy you will maximize the cumulative discounted occurrence. If you were to change your instance reward to $\mathbb{1} \\{(s_i, a_i) \in \cup_{j=t}^k \\{(s_j, a_j)\\}\\}$, i.e. the reward is 1 if you have reached the state action pair in the past, then you would solve the task faster, because in that case you don't care of staying in the state-action pair, but just going there as fast as possible.
- In line 280/281 you say that the agent will pay a cost of $-0.2$. Is the agent going to pay the price only when he pushes the button or in every step when the button is on?

**Questions:**

Look at Weaknesses section.

**Limitations:**

Limitations have been adequately addressed.

---

> ### Author Rebuttal · Authors · 2024-08-06
>
> Thank you for your helpful insights and suggestions. Below, we discuss the main points you raised.
>
> 1. We will make the distinction between Mon-MDPs and sparse-rewards MDP more explicit.
>
> 2. You are correct, the agent sees both states. We will make it clearer in the final version.
>
> 3. You are correct that, generally speaking, $\log(t)$ may not grow at a slower rate than $N_t(s,a)$. However, this holds in our case and that sentence was just an informal statement to give the reader an intuition of why our algorithm converges. Indeed, in line 175 right after that sentence we wrote “(formal proof below)”. We apologize for the confusion, and we will make it clearer in the final version.
>
> 4. Theorem 1 is formalized using the classic MDP notation for the sake of simplicity, and we now see how this may actually cause confusion. $Q(s,a)$ depends on both the monitor and the environment, i.e., its explicit notation for Mon-MDPs would be $Q(s_e, s_m, a_e, a_m)$.
> For Q-learning to converge in Mon-MDPs we need the following assumption: that the monitor is “truthful”, i.e., it either hides the reward or shows it as is (Parisi et al., Monitored Markov Decision Processes, 2023). Given a truthful monitor, under ergodicity and infinite exploration (proved in Theorem 1), then $\widehat Q$ converges to $Q^*$. We mentioned the notion of “truthful monitor” in footnote 2, page 3, but we will now make it explicit in Corollary 1. We apologize for the confusion.
>
> 5. Our reference to “as fast as possible” is imprecise, although not for the reason given, and we will clarify this in future revisions. The expected return from further visits to $s_i, a_j$ upon reaching the state-action pair under the optimal policy is a fixed positive value (as the environment is Markovian and rewards are all non-negative). Therefore, the optimal policy for accumulating visit counts to the state-action pair is still to visit the state action as quickly as possible, as this results in minimizing the discount factor on the visit reward and the fixed positive future return. And this is exactly what your proposed reward function would also do (just not include the fixed positive future return), while avoiding the non-Markovian property of your specification.
> However, the “as fast as possible” phrasing is imprecise because of how it handles stochasticity.  One might expect “as fast as possible” to minimize the expected time to reach the state-action pair, which the SR only does if the environment is deterministic. It is still incentivized to visit the state-action pair quickly, just with a different tradeoff between distributions of visit times.
>
> 6. Every step the button is on.

---

> > ### Comment · Reviewer_Y2Vd · 2024-08-11
> >
> > Thanks for your explanation! My concerns have been addressed. I will raise my score accordingly.

---

### Official Review · Reviewer_zrqd · 2024-07-08

**Soundness:** 2
**Presentation:** 3
**Contribution:** 3
**Rating:** 6
**Confidence:** 3

**Summary:**

This paper tackles the problem of exploration in MDPs where the reward is unobservable. For this, the authors perform goal conditioned exploration, i.e., the environment is explored according to how often a specified goal is reached.  The authors propose an exploitation-exploration mechanism based on learning goal-conditioned policies. Policies are switched as visitation counts increase for goal state-action pair for a give goal conditioned policy. Moreover, the authors propose learning separate value functions via successor features for each goal conditioned policy. Overall, this mitigates the unobserved reward problem since the reward is not crucial anymore to solving the environment.  In the environments proposed by the authors their approach outperforms other classic exploration methods especially when rewards unobserved, e.g., the reward is only given if the agent elicits a mechanism first.

**Strengths:**

- The paper is generally well written. I found the illustrations to be useful in understanding the ideas.
- The authors perform good experiments, describing baselines the baselines appropriately putting their method’s performance in to context.
- The idea of selecting goals systematically based on the count and then following a goal conditioned policy seems effective for the Mon-MDP setting.

**Weaknesses:**

- What makes the Mon-MDP different to a sparse rewards problem where a reward is only given at the end or reward free exploration for that matter? Shouldn’t your method be compared with other popular (intrinsic) exploration methods?
- Although the algorithm is effective in tabular spaces it seems non-trivial to expand it to continuous or high-dimensions. The author mention their intentions and cite papers but do not elaborate on for instance how to systematically choose goals in high-dimensional spaces.
- There seem to be no other Mon-MDP specific methods that have been compared against. Are there any other methods?

**Questions:**

Questions:
- What is the main contribution of the paper? Are the successor features crucial or is it the exploration-exploitation mechanism? The authors motivate the problem very well, but I feel it is not exactly explained why these two components are crucial to improved performance in Mon-MDPS
- The authors propose their own set of environments to test their method with success. Are there any other known environments that benefit from the paradigm of unobserved rewards?
- Why do the other methods struggle so much on the empty environment? Is it because the sparse reward problem is harder?

**Limitations:**

I think the authors have adequately addressed limitations in their work.

---

> ### Author Rebuttal · Authors · 2024-08-06
>
> Thank you for your helpful insights and suggestions. Below, we discuss the main points you raised.
>
> 1. In sparse-reward RL, rewards are mostly 0 with very few exceptions ("meaningful rewards"). In Mon-MDPs, the agent cannot see the rewards, not even the 0s. While intrinsic rewards may improve exploration, they cannot replace $r_e = \bot$ (unobservable), thus the agent cannot directly maximize the sum of environment rewards. There is a need for a mechanism to learn even in the absence of rewards. This is what Mon-MDPs are for, as introduced by *"Parisi et al., Monitored Markov Decision Processes, 2023"*.
> We also argue that most intrinsic rewards induce non-stationarity (e.g., counts change over time), and are myopic (counts rewards only immediate visits, not long-term visits).
> Since our evaluation is limited to discrete MDPs, we used intrinsic rewards based on counts (*"Bellemare et al, Unifying count-based exploration and intrinsic motivation, 2016"*). To the best of our knowledge, in fact, count-based rewards are the go-to choice for discrete MDPs, while other intrinsic-reward algorithms (e.g., RIDE by Raileanu and Rocktäschel, 2020; RND by Burda et al., 2018; Curiosity-driven by Pathak et al, 2017) are more suited for continuous MDPs and deep RL. In our follow-up on continuous Mon-MDPs, we will evaluate the latest deep RL intrinsic-reward algorithms.
>
> 2. We are currently working on extending our algorithm, as promised in Section 5. Here are more details that we will be happy to add to the paper as well. First, Universal Value Function Approximators (UVFA, Schaul et al., 2015) will replace the set of S-function. That is, while first we had one S-function for every state-action (each implemented as a table) now we have one single neural network that takes the goal state and the current state as input, and outputs the action value for each goal action. That is, $S(s_{goal}, s_t) = [a_{goal}, a_t]$.
> We have already implemented this and it works nicely, but we are still investigating UFVA (there are different versions, and we may even propose a novel one).
> Regarding the use of counts, the main challenge is the presence of $\arg\min N(s,a)$. To make it tractable, we plan to either use Random Network Distillation (RND, Burda et al., 2018), or a version of prioritized experience replay (Shaul, 2016) that takes into account either pseudocount or RND prediction errors (rather than TD errors).
>
> 3. There is indeed no other Mon-MDP method yet. Mon-MDPs were introduced very recently (Parisi et al, 2023) and ours is the first work addressing exploration in Mon-MDPs.
>
> 4. The main contribution is the explore-exploit paradigm that overcomes the limitations of optimism and therefore is effective for problems where rewards are unobservable. In order to make it work, SFs are a necessary component, as they allow to decouple exploration (SF: visit a desired state-action pair) from exploration (Q-function: maximize return).
> On one hand, Algorithm 1 formalizes how to decouple exploration and exploitation, and the mechanism to balance the two (i.e., the ratio $\beta_t$). Theorem 1 further proves its convergence. On the other hand, Algorithm 1 depicts a general approach that uses a generic “goal-conditioned policy $\rho$”. To make it practical, we propose S-functions. While using SFs as value functions is not novel in RL literature, we argue that the way we use them is.
>
> 5. We believe that Mon-MDPs better capture the complexity of real problems, and therefore any MDP can be extended to Mon-MDPs. In a follow-up work, we are modeling a roborace problem as Mon-MDP: the agent receives rewards for a trajectory only upon explicitly asking for it, and the availability of the feedback is modeled as a monitor. This relates to RL from human feedback, with the difference that the availability of the feedback follows a Markovian process and the agent can exploit it. The goal is that using Mon-MDP the agent can ask for feedback more effectively, and in the end will learn to race without never asking for it.
>
> 6. All algorithms (ours and baselines) learn a reward model to compensate for unobservable rewards: since the benchmark Mon-MDPs are ergodic, and because every environment reward is observable for at least one monitor state (e.g., when the button is on), given infinite exploration Q-learning is guaranteed to convergence (this was proven in “Parisi et al., Monitored Markov Decision Processes, 2023”).
> The problem with the baselines (not ours) is that they use the Q-function to explore. In $\epsilon$-greedy (red) and intrinsic reward (green), the greedy operator is over Q; UCB-like (orange) still considers Q in its greedy operator; optimism (black) is pure greedy over Q.
> This is a problem in Mon-MDP because Q-function updates are based on the reward model that is inaccurate at the beginning (if the reward is unobservable, the agent queries the reward model). To learn the reward model and produce accurate updates the agent must perform suboptimal actions and observe rewards. This creates a vicious cycle in exploration strategies that rely on the Q-function: the agent must explore to learn the reward model, but the Q-function will mislead the agent and provide unreliable exploration (especially if optimistic).
> Our algorithm, instead, builds its exploration over the S-function, whose reward is always observable (either 1 or 0). This allows the agent to quickly learn accurate S-functions, which will then produce efficient and uniform exploration (as much as possible, at least, since visitation also depends on the initial state and the transition function). By visiting all environment and monitor states efficiently, the agent can also observe environment rewards quickly, learn an accurate reward model, and finally the optimal Q-function.
> We will write a dedicated paragraph to this explanation in the final version.

---

> ### Comment · Reviewer_zrqd · 2024-08-11
>
> I thank the authors for their exhaustive answer!
>
> * I understand the relevance of Mon-MDPs and how they differ from sparse-rewards exploration much better now.
>
>
> * I share the same concern as reviewer DPys, where all other baselines tested are not aware of the formalism and thus naturally perform worse.
> * Also from reading the paper it does not become apparent to me why the goal directed exploration with successor representations is the best way to address the Mon-MDP formalism? I think this has to be very clear, since this seems to be the first concise method to address Mon-MDPs.
>
> * Could you maybe elaborate on this? I think this would help me gain more confidence in the paper.
>
>
> Thank you!

---

> > ### Author Response · Authors · 2024-08-11
> >
> > The fact that other baselines are not aware of the formalism highlights a hole in RL literature, i.e., the lack of algorithms for MDPs with partially-observable rewards (that are not trivial, e.g., the observability of rewards is not just binary but given by another MDP).
> > While there exists reward-free algorithms, they split exploration and exploitation: first, a general exploration policy is learned (environment rewards do not exists at that time), and then tasks are learned. In our paper, we are interested in solving a given task for which rewards **do exists** but are not observable --- the agent is still being evaluated even though it cannot see (sometimes) the evaluation.
> > We argue that this is an important hole in RL literature, and we discussed more on this in the general rebuttal.
> > In particular, a major advantage of our algorithm is indeed not having two separate exploration/exploitation phases: even if sub-optimal at firtst, the goal-conditioned policies are still better visitation policies than classic ones.
> > In this paper, we strived for simplicify and efficiency when we designed our algorithm. The main contribution is to highlight the failure of optimism in Mon-MDPs and the introduction of the general explore-exploit algorithm.
> >
> >
> > SFs naturally satisfy the requirements of our goal-conditioned policy $\rho$ (Section 3, paragraph after Corollary 1) while being simple and straightfoward to implement. A policy maximizing the SFs will maximize the visitation of state-action pairs, while being completely independent from the Q-function (that, as discussed in our previous reply, can be highly inaccurate at the beginning).
> > We don't claim this is the **best** way to address Mon-MPDs but certainly is effective, and it is a first step to fill the hole in RL literature.

---

### Official Review · Reviewer_ajhV · 2024-07-08

**Soundness:** 3
**Presentation:** 3
**Contribution:** 3
**Rating:** 6
**Confidence:** 5

**Summary:**

The authors propose a novel exploration strategy for Mon-MDPs based on two policies; a goal-conditioned exploration policy and an exploitation policy which maximizes the underlying reward. The proposed algorithm alternates between the two policies, naturally trading off exploration and exploitation.
They show that the proposed strategy is consistent and outperforms standard exploration approaches such as optimistic exploration.

**Strengths:**

The paper is very well written. Particularly, the analysis of explore and exploit strategies for Mon-MDPs is interesting and novel. The empirical results are also good.

**Weaknesses:**

There are works in reward-free RL [1, 2] and active learning in RL [3, 5] for general continuous state-action spaces that the authors should mention. Since their strategy is based on an explore and exploit approach, intuitively, it seems that works such as [4] can be seamlessly applied to the Mon-MDP setup. Also, from my understanding, the intrinsic reward proposed in [4] is not myopic and is also consistent, in the sense that it leads to convergence guarantees for learning the MDP. Hence, I am not sure I agree with the authors' statement on lines 154 -- 155.

There are also works on goal-conditioned RL such as [5, 6] that build on a similar idea for picking and exploring novel goals for exploration. While they do not consider the Mon-MDP setting, I think they are still somewhat relevant to prior work.

The authors should write the rate of convergence in the main theorem statement of Theorem 1.

[1] Jin, Chi, et al. "Reward-free exploration for reinforcement learning." International Conference on Machine Learning. PMLR, 2020.
[2] Chen, Jinglin, et al. "On the statistical efficiency of reward-free exploration in non-linear rl." Advances in Neural Information Processing Systems 35 (2022): 20960-20973.
[3] Mania, Horia, Michael I. Jordan, and Benjamin Recht. "Active learning for nonlinear system identification with guarantees." arXiv preprint arXiv:2006.10277 (2020).
[4] Sukhija, Bhavya et al. "Optimistic active exploration of dynamical systems." Advances in Neural Information Processing Systems 36 (2023): 38122-38153.
[5] Nair, Ashvin V., et al. "Visual reinforcement learning with imagined goals." Advances in neural information processing systems 31 (2018).
[6] Hu, Edward S., et al. "Planning goals for exploration." arXiv preprint arXiv:2303.13002 (2023).

**Questions:**

1. Instead of a goal-conditioned policy, could one not train a policy with $1/N_t(s, a)$ as reward (or $-N_t(s, a)$)? In essence, this policy will try to visit states that have a low visitation count. While the reward in this setting is non-stationary, empirically this might perform better. What do the authors think about it?

2. Could [4] from above be used for active learning of dynamics and then later exploitation for general continuous state-action spaces with Mon-MDPs? Moreover, I am curious if any general active learning/reward-free RL algorithm can be combined with greedy exploitation to obtain convergence guarantees for Mon-MDPs.


I am happy to increase my score if my concerns and questions above are adequately addressed.

**Limitations:**

The authors address the limitations of their work in the main paper (lines 336  -- 349).

---

> ### Author Rebuttal · Authors · 2024-08-06
>
> Thank you for your helpful insights and suggestions. Below, we discuss the main points you raised.
>
> 1. Thank you for the additional references, we will add them to the final version.
> In particular, we think that [1] is indeed close to our approach in the use of SF. It is different, however, as it has two separate stages for exploration and exploitation, while ours balances between the two using the coefficient $\beta_t$ (Algorithm 1, line 2).
> We would like to stress, however, that [3], [4], and [6] propose model-based algorithms.
> In our paper, we focused on model-free RL and thus evaluated our algorithms against model-free baselines. As discussed in Section 5, we will devote future work on model-based versions of our algorithm.
>
> 2. We don't have a rate of convergence for Theorem 1, only an asymptotic convergence proof. Evidence for the practicality, in terms of sample efficiency, of our proposed algorithm is instead given by our thorough empirical experiments. Provable rates of convergence often don't imply practical algorithms (e.g., [1] and [2], which don't include any experimental results), which was one of the goals for our paper.
>
> 3. We agree that this is an interesting baseline, and we will add it to the final version.
>
> 4. Active RL (ARL) is perhaps the closest framework to Mon-MDPs but its setting is simpler. To the best of our knowledge, ARL considers only binary actions to request rewards, constant request costs, and perfect reward observations. By contrast, in Mon-MDPs (a) the observed reward depends on the monitor — a process with its own states, actions, and dynamics; (b) there may be no direct action to request rewards, and requests may fail; (c) the monitor reward is not necessarily a cost. For these reasons, ARL can be seen as a special case of Mon-MDPs, and therefore cannot fully capture the complexity of Mon-MDPs.

---

> > ### Comment · Reviewer_ajhV · 2024-08-07
> > **Response to Author's rebuttal**
> >
> > 2. I think not having convergence rates is a drawback of the work. What is it that limits the authors from doing so? Intuitively, wouldn't a rate on the visitation frequency directly result in a rate for Theorem 1? This is what is at least shown in [4] (c.f., Lemma 13)
> >
> > 4. I am not sure if this is true for methods such as [3, 4]. In the end, they are similar in spirit to the proposed algorithm -- they visit the states where they have the highest uncertainty/lowest visitation count (exploration phase). In principle, they could be combined with a similar exploitation phase as proposed by the authors. Am I missing something here? In the end, this approach would be equivalent to the baseline discussed in the third point. Whereas, the algorithm in [6] is similar to what the authors propose.
> > Could the authors comment more on this?

---

> > > ### Author Response · Authors · 2024-08-08
> > > **About convergence rate and related work**
> > >
> > > **About the convergence rate**
> > >
> > > Getting a non-trivial convergence rate is likely not possible. The algorithm as presented uses $\epsilon$-greedy as the exploration method to learn the goal-conditioned S-functions. In the worst-case, this will admit $\epsilon$-greedy's poor convergence rate (and may take exponential-time in MDP size to even just visit the goal once). We could give a competitive convergence rate if optimal goal-conditioned S-functions are known in advance, but that didn't feel particularly informative.
> > > We could also replace the exploration mechanism (i.e., to follow $\rho$ and S-functions) within the exploration phase to use a more sophisticated mechanism, but we are explicitly aiming for a model-free algorithm so many choices common in literature are not suitable (e.g. MBIE, UCRL).
> > > For example, in [4], Lemma 13 refers to Eq. 16 where the problem considers the true transition function $f^*$, and the make use of it in Corollary 7. On the contrary, we don't learn any model, and we don't know the true S-function $S^*$ either. Even knowing $S^*$, we don't think Lemma 13 can be straightforwardly applied to our case.
> > >
> > > We choose a model-free algorithm (Q-learning) because we wanted to keep the method as simple as possible, while demonstrating its effectiveness through experimentation rather than theory, particularly showing that it can handle partially observable reward settings like Mon-MDPs (where traditional MDP exploration mechanisms can fail).
> > > Maybe a middle-ground might be to present a convergence rate that implicitly depends on the convergence rate of whatever algorithm is used to learn the goal-conditioned S-functions. This could show how much is lost due to the goal-directed exploration component that is needed to handle Mon-MDPs. We would then continue with experiments showing its performance with the simple $\epsilon$-greedy mechanism (even though it's not theoretically well-motivated) showing its strong performance across the tested environments.
> > >
> > > There is another angle of analysis that is distinctly more difficult as it likely would need new theoretical machinery (beyond the scope of this work). In particularly, we believe that most of the advantage of our algorithm comes from the goal-conditioned policies finding sub-optimal but still useful visitation policies. These policies can still be used to dramatically accelerate the learning of the Q-functions, allowing fast exploitation possibly even before the S-functions have identified optimal exploration policies $\rho$.
> > > We believe that this is a great advantage of not breaking the problem into some explicit exploration-only phase (e.g., as in [1]). This would almost certainly need some instance-specific analysis, though.
> > >
> > > Would the middle-ground option of a convergence rate that depended on the convergence rate of the goal-conditioned policy learning be a sufficient addition? In any case, we can add a more thorough discussion of this in the paper.
> > >
> > >
> > >
> > > **About related work**
> > >
> > > Yes, they could be combined with our approach. Maybe there has been some misunderstanding: we didn't mean that [3], [4], and [6] are unrelated, but just that they are different being model-based.
> > >
> > > For example, [3] estimates uncertainty in the feature space and uses a model to plan a trajectory to high-uncertainty states. We use counts (rather than uncertainty) and don't plan trajectory, but instead follow one-step S-function values. While replacing uncertainty with counts could be straightforward, [3] still has an extra component (the model) that allows for more powerful long-term reasoning.
> > > We believe that combining our algorithm with [3] and [4] could be promising for future work.
> > >
> > > Regarding [6], the main similarity is the presence of a goal-conditioned policy for exploration, but their algorithm is still quite different (beside being model-based).
> > > First, [6] alternates between exploration and exploitation at every episode, while ours uses a more grounded criterion (the ratio $\beta).
> > > Second, the goal of their goal-conditioned policy is randomly sampled from the replay buffer and optimized via MPPI, while ours is the state-action pair with the lowest count.
> > > Third, their reward for training the goal-conditioned policy is given only at the end of the trajectory and depends on the number of actions needed to reach the goal, while ours is given at every step by SRs.
> > >
> > > We'd like to stress once more, however, that we didn't mean to say that [3], [4], and [6] are unrelated, and we will reference them in the final version.

---

> > > > ### Comment · Reviewer_ajhV · 2024-08-09
> > > > **Response to Author's rebuttal**
> > > >
> > > > Thanks, my concerns are clarified. While I would like to stick to my score for the paper, I have gained more confidence in the paper, i.e., I have increased my confidence from 3 to 5.

---

### Official Review · Reviewer_DPys · 2024-07-12

**Soundness:** 3
**Presentation:** 4
**Contribution:** 2
**Rating:** 6
**Confidence:** 4

**Summary:**

This paper considers the Reinforcement Learning problem in Monitored MDPs. Monitored MDPs are a formalism that has been recently introduced by Parisi et al. (AAMAS 2024), in which the value of the policy is computed on the rewards generated by the environment and those produced by a "monitor". However, the agent cannot directly observe the rewards generated by the environment, because the monitor can modify the reward observed by the agent. The authors propose an algorithm for Monitored MDPs, which alternates explicit exploration and exploitation. The method has been validated against some classic RL algorithms and variants.

**Strengths:**

(quality) The paper is very well-written. The topics are presented very clearly and the text can be read from top to bottom whithout any major misunderstanding. The related work section covers the essential work in the area (at least, the ones I am aware of), and the papers are described well. The main contribution is motivated and it is described completely.

(relevance) The study of Monitored MDPs has important relations with partially observable environments, and it is very relevant for the broad RL community. Moreover, it clarifies the difference between observing rewards and being only evaluated on them.

(reproducibility) The authors provided a full, well-documented Python source code of the algorithm, which ensures reproducibility. They also provided the necessary configuration files.

(soundness) The proposed algorithm appears to be appropriate for the class of domains and the monitors considered in the experimental section. Also, the evaluation considers 5 monitors for each of the 4 environments, which is an interesting composition.

**Weaknesses:**

1- Corollary 1 is the only formal guarantee regarding the performance of the algorithm that has been obtained in the paper. However, this only involves the asympotic convergence. Moreover, the proof relies on the fact that the algorithm is an exploratory policy that becomes greedy in the limit. This fact does not seem to suffice for obtaining $\hat{Q} \to Q^*$, because the monitor may arbitrarily modify the reward used for constructing $\hat{Q}$. I believe there is an implicit (but missing) assumption that the monitor may only hide rewards, and that rewards will be shown in all states an infinite number of times.

2- The evaluation compares the algorithm with other 4 baselines in some environments. However, none of the baselines have been specifically designed for Monitored MDP, nor for any component of partial observability on rewards, and it is unsurprising that many of these do not perform well. As such,the experimental evaluation gives little insights about the improvement that the proposed algorithm leads among Monitored MDP algorithms. If RL algorithms for this class do not exist yet, because the formalism is recent, then it should be also compared with algorithms are  aware that rewards are partially observable, or nonstationary. Finally, the black line, which is Q-learning with optimistic initialization, is taken as a representative of "optimism". However, this is only a specific instance of optimistic approaches, and I believe that "optimism" would be better represented by UCB-style RL algorithms.

3- The contribution of the paper is limited for the following reasons:

  a- The algorithm mostly succeeds because it tries to reach all states of the MDPs via explicit exploration. This is generally intractable for large state spaces, or MDPs with high (or infinite) diameter. Optimism solves this issue by avoiding exhaustive exploration. This makes the idea behind the algorithm more naive than other existing optimistic approaches, especially for larger state spaces or small probabilities. Indeed, the algorithm does not seem to directly address the fact that a monitor is present, even though it has this prior information available.

  b- The use of Successor Representation made in the paper is mostly standard in the literature. The authors say that only Machado et al. [40] used SR to drive exploration. However, we should consider that using SR as value functions is completely equivalent to placing unit rewards in goal states. Then, similar approaches are taken by the works in goal-conditioned RL that learn goals using unit rewards, even though SR are not explicitly mentioned in those papers.

**Questions:**

4- Why does the UCB baseline use an epsilon-greedy policy? I would expect bonuses to be sufficient for exploration. Epsilon-greedy may unnecessarily show it down.

The authors may also address any of the weaknesses above, especially number 1.

**Limitations:**

Most limitations have been already discussed in the paper. The work has no direct societal impact.

---

> ### Author Rebuttal · Authors · 2024-08-06
>
> Thank you for your helpful insights and suggestions. Below, we discuss the main points you raised.
>
> 1. You are correct, we consider only "truthful monitors" (as formalized in *"Parisi et al., “Monitored Markov Decision Processes, 2023”*), i.e., monitors that either hide the reward or show it as is. We wrote in page 3, footnote 2 that *"the monitor does not alter the environment reward"*. We will make it more explicit in Corollary 1.
> This is needed for our proof of convergence. The investigation of monitors that can change the reward are out of the scope of this paper and relates other directions of research (e.g., *"Ng et al., Policy invariance under reward transformations: Theory and application to reward shaping, 1999”*). Relaxing this condition, in fact, poses an additional challenge that could be addressed by having a belief over the reward (e.g., *"Marom and Rosman. Belief reward shaping in reinforcement learning, 2018"*).
>
> 2. You are correct that no baselines exist yet for Mon-MDPs because the formalism is recent. For a fair comparison against existing baselines, we included standard MDPs in our evaluation ("Full Observ." environments in Figures 5 and 6). The results show that even in classic MDPs our approach works better than existing ones: most baselines converge but they need a significantly larger amount of data. In Mon-MDPs, only ours converges in most seeds and the baselines fail most of the time.
> We didn't include UCB-based RL algorithms like UCRL because they are model-based and we considered only model-free methods for a fair comparison. Model-free alternatives (e.g., *“Dong et al., Q-learning with UCB exploration is sample efficient, 2020”*) usually provide guarantees of convergence (with convergence rates) but are impractical and lack evaluation on even simple domains.
> Also, we included a UCB-like baseline (orange line in Figure 5 and 6) that uses the UCB bonus to encourage exploration.
> We are happy to include more baselines if you have any suggestions.
>
> 3. a) We agree and we already discussed extending our algorithm to larger/continuous Mon-MDP and to model-based approach (e.g., learning the monitor model) in Section 5. We are currently working on extending it to larger/continuous domains using UVFA and alternative to counts, and we plan to submit a follow-up soon. Please refer to the response to reviewer **zrqd** for more details.
> We argue, however, that these limitations should not overshadow the contribution of the paper. We believe that the whole Mon-MDP framework is still very new and unexplored, and that our approach, proof of convergence, and evaluation are novel and thorough enough, and set the stage for many future directions of research.
>
> 3. b) Thank you for this insight. How would it be equivalent to place unit rewards in goal states, though? In our algorithm, one SF places unit rewards in **one** state, and then we learn a SF for **every** state. We see similarities between the classic use of SF and our approach, but usually SF are given an explicit goal (or features representing it) and are applied in the context of transfer learning. In our case, there are as many goals as state-action pairs.
> This is why we wrote that only Machado et al. applied SF for exploration, since their work was tailored to encourage visitation of state-action pairs.
> We are happy to cite more related work if you have any suggestions.
>
> 4. Greedy UCB performed better in some environments but very poorly in others. The average performance was best with the addition of $\epsilon$-greedy action-selection.

---

> > ### Comment · Reviewer_DPys · 2024-08-12
> >
> > 1. I appreciate this important change. This assumption is very reasonable to assume and it would not limit the impact of the paper.
> >
> > 4. b) I said that this use of successor representations is mostly standard in goal-conditioned RL literature because from what concerns the policy $\rho$, the approach appears to be equivalent to learning $|SA|$ policies using Q-lerning, where a reward of 1 is placed in one state-action pair for each policy. The overall algorithm is still original, to the best of my knowledge. I mainly argue that the exploration strategy is not only related to directed exploration with SR features, but also goal-conditioned policies.
> >
> > The other replies are also relevant.
> > My main remaining concern is that the technique "is generally intractable for large state spaces, or MDPs with high (or infinite) diameter.", given that the algorithm must learn a distinct policy for each state and action in the MDP.
> >
> > Despite this significant limitation, the paper is well written, clear, correct and the results appear to be mostly reproducible. So, I have increased my evaluation to weak accept.

---

### Author Rebuttal · Authors · 2024-08-06

We thank the reviewers for their feedback and helpful suggestions. We are pleased to see that all reviewers appreciate the core idea of our paper — a novel exploration algorithm for Mon-MDPs, where rewards are partially observable — and the thorough evaluation against different baselines on many environments/monitors. We are also happy to see that all reviewers find our paper intuitive and very well-written.

We have replied to each reviewer separately to address some specific clarifications. In this comment, we recap what we will introduce in the final version of the paper after taking into account the reviews.

* We will make it clearer why Mon-MDPs are different from sparse-rewards MDPs (**zrqd** and **Y2Vd**).
* We will add more related work (**ajhV**).
* We will clarify that the agent sees both the environment and the monitor state (**Y2Vd**).
* We will make it more explicit that convergence is guaranteed under a “truthful monitor” (**DPys** and **Y2Vd**).
* Will will rephrase why S-functions lead the agent to the goal state “as fast as possible” (**zrqd**).
* We will add another baseline that learns a separate Q-function with intrinsic reward $1 / N(s,a)$ (**ajhV**).
* We will discuss more why Mon-MDPs are so challenging and why all baselines perform poorly (**zrqd**).
* We will discuss more about future work and how we plan to extend our algorithm to continuous spaces (**DPys** and **zrqd**).

We further thank reviewer **DPys** for saying that *"the main contribution is motivated and it is described completely”* and *“the study of Monitored MDPs has important relations with partially observable environments, and it is very relevant for the broad RL community. Moreover, it clarifies the difference between observing rewards and being only evaluated on them”*.
We would like to follow-up on this, and emphasize that our paper highlights an important hole in the RL literature: MDPs assume rewards are observable, and areas of research that investigate cases where rewards are not observable (e.g., Active RL) still consider limiting assumptions. Mon-MDPs do not place limitations and formalize the observability of rewards as a separate Markov process, giving great freedom on how to model real-world problems.
At the same time, however, Mon-MDPs highlight the lack of efficient algorithms with guarantees of convergence and the failure of optimism. While this is well-known for bandits (*“Lattimore and Szepesvari, The end of optimism? An asymptotic analysis of finite-armed linear bandits, 2017”*), RL literature lacks a similar analysis on MDPs. With this paper, we want to contribute to filling this gap with the introduction of an algorithm with guarantees of convergence and a collection of benchmarks, and set the stage for more exciting future research.

---

### Decision · Program_Chairs · 2024-09-25

**Decision:**

Accept (poster)

**Comment:**

The paper studies a recently introduced class of sequential decision problems called Monitored MDPs, and argues that the principle of optimism in the face of uncertainty leads to sub-optimal exploration when the rewards are unobservable. The authors show instead that a goal-conditioned exploration strategy can resolve uncertainty correctly even when rewards are unobserved, and show empirically across RL benchmarks (including new ones with partially observable rewards) that the proposed techniques outperform standard RL baselines.

The reviewers raised several clarifying questions, that the authors answered convincingly in their rebuttal. Including the discussions in the paper revision will substantially strengthen the paper. There is some uncertainty whether an additional baseline (separate Q-functions trained for a different intrinsic reward) will substantially change the conclusions from the experiments; but all the reviewers agree that nevertheless the paper's theoretical contributions to our understanding of Monitored MDPs (and partial monitoring bandits) are likely to be of independent interest to the RL community.